# SMPDL3b modulates insulin receptor signaling in diabetic kidney disease

A. Mitrofanova[1,2,3], S.K. Mallela[1,2], G.M. Ducasa[1,2,4], T.H. Yoo[1,2,5], E. Rosenfeld-Gur[6], I.D. Zelnik[6], J. Molina[1,2], J. Varona Santos[1,2], M. Ge[1,2,4], A. Sloan[1,2], J.J. Kim[1,2], C. Pedigo[1,2,7], J. Bryn[1,2], I. Volosenco [1,2,8], C. Faul[1,2,9], Y. H. Zeidan[2,10,11], C. Garcia Hernandez[1,2,10], A.J. Mendez[12], I. Leibiger[13], G.W. Burke[3,12], A.H. Futerman[6], L. Barisoni[14], Y. Ishimoto [15,16], R. Inagi [15,16], S. Merscher[1,2] & A. Fornoni[1,2]

Sphingomyelin phosphodiesterase acid-like 3b (SMPDL3b) is a lipid raft enzyme that regulates plasma membrane (PM) fluidity. Here we report that SMPDL3b excess, as observed in podocytes in diabetic kidney disease (DKD), impairs insulin receptor isoform B-dependent pro-survival insulin signaling by interfering with insulin receptor isoforms binding to caveolin-1 in the PM. SMPDL3b excess affects the production of active sphingolipids resulting in decreased ceramide-1-phosphate (C1P) content as observed in human podocytes in vitro and in kidney cortexes of diabetic db/db mice in vivo. Podocyte-specific *Smpdl3b* deficiency in db/db mice is sufficient to restore kidney cortex C1P content and to protect from DKD. Exogenous administration of C1P restores IR signaling in vitro and prevents established DKD progression in vivo. Taken together, we identify SMPDL3b as a modulator of insulin signaling and demonstrate that supplementation with exogenous C1P may represent a lipid therapeutic strategy to treat diabetic complications such as DKD.

[1] Katz Family Division of Nephrology and Hypertension, Department of Medicine, University of Miami, Miller School of Medicine, Miami 33136 FL, USA. [2] Peggy and Harold Katz Family Drug Discovery Center, University of Miami, Miller School of Medicine, Miami 33136 FL, USA. [3] Department of Surgery, University of Miami, Miller School of Medicine, Miami 33136 FL, USA. [4] Department of Molecular and Cellular Pharmacology, University of Miami, Miller School of Medicine, Miami 33136 FL, USA. [5] Department of Internal Medicine, College of Medicine, Yonsei University, Seoul 03722, Korea. [6] Department of Biological Chemistry, Weizmann Institute of Science, Rehovot 76100, Israel. [7] Department of Internal Medicine, Yale University School of Medicine, New Haven 06510 CT, USA. [8] Lewis Gale Medical Center, Salem 24153 VI, USA. [9] Division of Nephrology, Department of Medicine, University of Alabama at Birmingham, Birmingham 35233 AL, USA. [10] Department of Radiation Oncology, University of Miami, Miller School of Medicine, Miami 33136 FL, USA. [11] Department of Radiation Oncology, American University of Beirut, Beirut 1107 2020, Lebanon. [12] Diabetes Research Institute, University of Miami, Miller School of Medicine, Miami 33136 FL, USA. [13] The Rolf Luft Research Center for Diabetes and Endocrinology, Karolinska Institutet, Stockholm 17176, Sweden. [14] Department of Pathology, University of Miami, Miller School of Medicine, Miami 33136 FL, USA. [15] Division of Nephrology and Endocrinology, University of Tokyo Graduate School of Medicine, Tokyo 113-8654, Japan. [16] Division of CKD Pathophysiology, University of Tokyo Graduate School of Medicine, Tokyo 113-8654, Japan. Correspondence and requests for materials should be addressed to A.F. (email: afornoni@med.miami.edu)

Sphingolipids and cholesterol are critical components of the eukaryotic plasma membrane (PM), where they serve as major mediators of lipid-signaling regulating multiple cellular functions[1]. We have identified sphingomyelin phosphodiesterase acid-like 3b (SMPDL3b) as a lipid raft enzyme that regulates integrin activation, cell migration, and cell survival in podocytes, which are terminally differentiated and specialized cells of the kidney glomerulus[2,3]. Others have demonstrated an important role of SMPDL3b in affecting membrane lipid composition and fluidity in macrophages[4]. Although the crystal structure of SMPDL3b was recently reported[5], the specificity of its phosphodiesterase activity and the possibility of other lipid modifying functions of SMPDL3b have not been investigated yet.

Ceramide is the centerpiece of the sphingolipid metabolic pathway, where it serves as substrate for the generation of ceramide-1-phosphate (C1P) and sphingosine 1-phosphate (S1P)[6]. C1P and S1P are bioactive sphingolipids regulating important processes, such as cell survival and inflammatory responses. S1P can be cleaved by S1P lyase, and S1P excess due to a genetic deficiency of S1P lyase was recently found to be associated with adrenal insufficiency, nephrotic syndrome, and ichthyosis[7–9]. In contrast, while reduction of biologically active C1P has been linked to apoptosis and altered inflammatory signaling, little is known about the metabolic or signaling pathways that are controlled by C1P and that regulate C1P content.

The overall goal of this study is to determine if SMPDL3b may affect the availability of biologically active lipids and alter the assembly of protein complexes in lipid raft domains thus interfering with physiological signaling. Among signaling complexes located in raft domains that are heavily dependent on the PM lipid composition, we are focused on the insulin receptor (IR)-signaling complex for several reasons: (1) IR signaling is a key modulator of podocyte function[10–12]; (2) SMPDL3b is strongly upregulated in glomeruli of patients with insulin resistance and diabetic kidney disease (DKD)[3]; (3) IR exists in two isoforms which are characterized by distinct affinities for lipid raft domains[13] and which are differentially expressed in various cell types suggesting distinct functions. We therefore tested the hypothesis that SMPDL3b affects the generation of sphingolipids involved in the regulation of IR signaling.

Terminally differentiated human podocytes are utilized as a model system to study in vitro whether SMPDL3b differentially affects IR isoform A (IRA) or IR isoform B (IRB) complex formation with caveolin-1 and IR-dependent signaling. A diabetic db/db mouse model with podocyte-specific deletion of Smpdl3b is used to determine if SMPDL3b deficiency protects from experimental DKD. Here we report and discuss that increased expression of SMPDL3b is associated with deficiency of biologically active C1P in vitro and in vivo. Exogenous C1P supplementation is sufficient to restore insulin signaling in SMPDL3b overexpressing (SMP OE) cells in vitro and to improve proteinuria in DKD in vivo. Our findings shed some light on the complexity of IR signaling, demonstrate an important role of SMPDL3b in the modulation of insulin signaling, and provide the first evidence that biologically active lipids, such as C1P may represent treatment options for complications of diabetes associated with high cardiovascular morbidity and mortality such as DKD.

## Results

**SMPDL3b affects the levels of C1P.** We previously described SMPDL3b as an enzyme expressed in podocyte lipid rafts that might play an important role in the pathogenesis of proteinuric kidney diseases[2,3]. Others showed that SMPDL3b functions as a phosphodiesterase with lipid-modifying properties that negatively regulates immunity in macrophages[4]. We first aimed at determining if SMPDL3b also acts as a phosphodiesterase in human podocytes and found that overexpression of SMPDL3b in podocytes (SMP OE) is associated with increased cellular phosphodiesterase activity, while knockdown of SMPDL3b (siSMP) resulted in decreased phosphodiesterase activity (Fig. 1a). In addition to this, the degree of SMPDL3b expression affected the content of active sphingolipids in podocytes. While the total sphingomyelin (Fig. 1b) and total ceramide (Fig. 1c) levels remained unchanged, total C1P levels were significantly decreased in SMP OE podocytes and increased in siSMP podocytes (Fig. 1d) compared to control podocytes (CTRL). To determine which ceramide species mostly contribute to the observed changes in C1P levels, liquid chromatography mass spectrometry (LC–MS) was performed. While human podocytes mostly express C16:0, C22:0, C24:0, and C24:1 ceramide and C16:0, C18:0, and C24:0 C1P species, only C16:0 C1P levels were significantly reduced in SMP OE and increased in siSMP podocytes (Supplementary Data 1).

In addition to its phosphodiesterase activity, protein sequence analysis of SMPDL3b using InterPro protein sequencing analysis tool, predicted the presence of a phosphatase-like domain. In order to understand if SMPDL3b may contribute to the decreased C1P content by converting C1P into ceramide, we utilized an in vitro assay in HEK293 cells transfected with full-length human SMPDL3b and found a significantly higher conversion of C1P to ceramide upon SMPDL3b overexpression (Fig. 1e). However, in vitro enzyme assay determination of SMPDL3b activity with pure recombinant protein failed to detect phosphatase activity (Supplementary Fig. 1). While this could be due to the utilization of a soluble non-GPI-anchored protein or to the absence of an essential cofactor, we can only conclude that SMPDL3b contributes to the conversion of C1P into ceramide.

**SMPDL3b affects IR signaling.** We previously observed that SMPDL3b is strongly upregulated in glomeruli of patients with insulin resistance and DKD[3]. This observation, together with the finding that preservation of insulin signaling is a key to protect podocytes from injury[10,12,14,15], prompted us to determine if SMPDL3b may affect insulin signaling pathways. mRNA sequence analysis in control and SMP OE podocytes was performed. We found differential expression of 1948 genes that passed FDR correction in SMP OE podocytes compared to control (Supplementary Data 2). Using DAVID pathway-enrichment analysis, we found that 62 signaling pathways are affected in SMPDL3b overexpression cells (Supplementary Data 3). Among them, differential expression of genes involved in the regulation of phosphoinositide 3-kinase and protein kinase B (PI3K-AKT), sphingolipid signaling, mammalian target of rapamycin (mTOR), and insulin resistance pathways was observed (Supplementary Data 3). In addition, similar to what was described in macrophages[4], differential regulation of genes important in innate immunity, involved in the tumor necrosis factor (TNF), nuclear factor kappa-light-chain-enhancer of activated B-cells (NFκB) and toll-like receptors (TLRs)-signaling pathways, was observed (Supplementary Data 3).

We first validated mRNAseq findings at the protein level, demonstrating that SMP OE cells, unlike siSMP or CTRL podocytes, are unable to phosphorylate protein kinase B (AKT), but become responsive to phosphorylate p70S6 kinase (p70S6K) (Fig. 2a, b) after stimulation with insulin. SMP OE podocytes also demonstrated increased phosphorylation of eukaryotic translation initiation factor 4E-binding protein-1 (4EB-P1), a downstream target of mTOR signaling, while the phosphorylation of 4EB-P1 was suppressed in siSMP and CTRL podocytes (Fig. 2a, b). Additionally, kinetics of phosphorylation of AKT, p70S6K, or

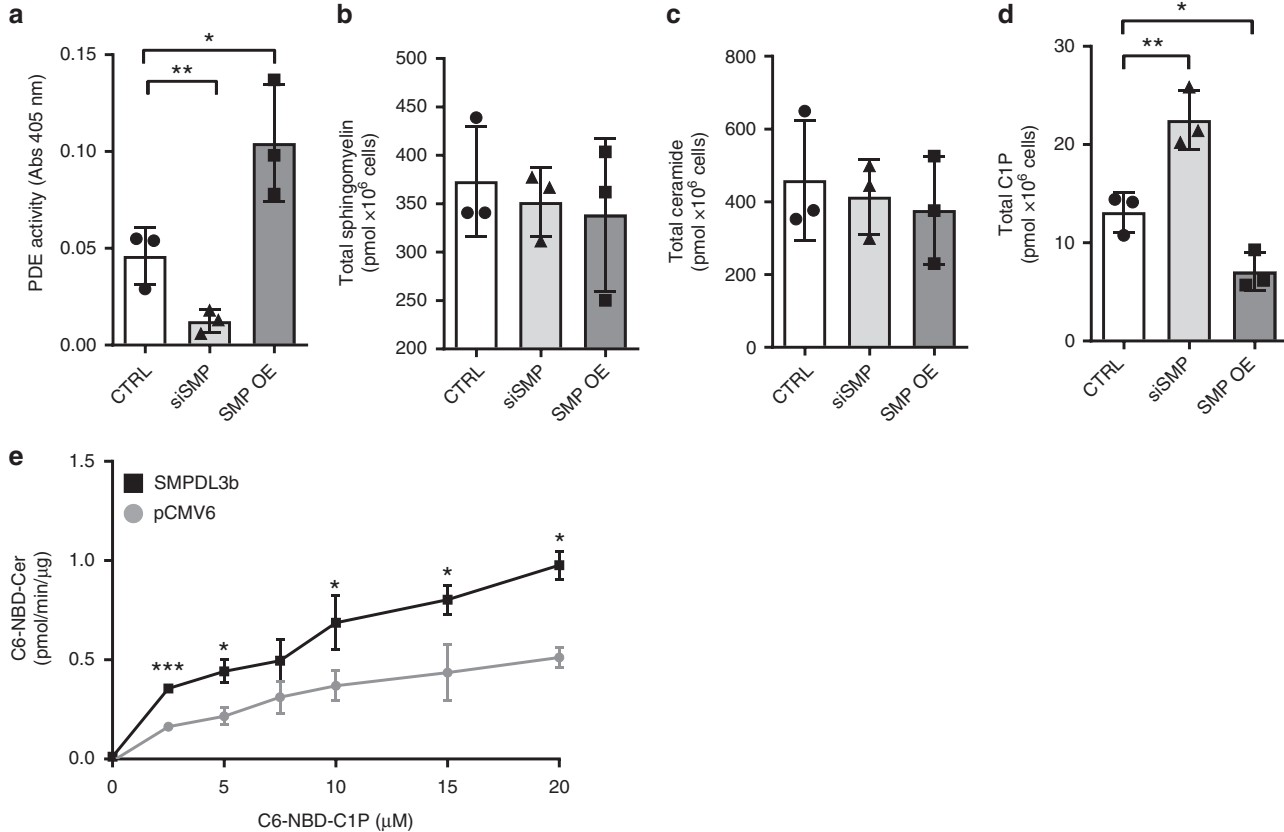

**Fig. 1** SMPDL3b affects the conversion of C1P to ceramide. **a** Bar graph analysis of phosphodiesterase (PDE) activity in control (CTRL), SMPDL3b knockdown (siSMP) and SMPDL3b overexpressing (SMP OE) human podocytes. $n = 3$ independent experiments; $*P = 0.039$, two-tailed $t$-test. **b** Liquid chromatography mass spectrometry analysis (LC–MS) of total sphingomyelin content in CTRL, siSMP, and SMP OE human podocytes. $n = 3$ independent experiments. $P = 0.78$, $F = 0.26$, one-way ANOVA. **c** LC–MS of total ceramide content in CTRL, siSMP, and SMP OE human podocytes. $n = 3$ independent experiments. $P = 0.78$, $F = 0.78$, one-way ANOVA. **d** LC–MS of total ceramide-1-phosphate (C1P) content in CTRL, siSMP, and SMP OE human podocytes. $n = 3$ independent experiments. $*P = 0.045$, $**P = 0.007$, $F = 32.58$, one-way ANOVA. **e** C1P phosphatase in vitro assay using increasing amounts C6-NBD-C1P. $n = 2$ independent experiments in duplicate; $*P < 0.05$, $***P < 0.001$; two-tailed $t$-test. Error bars represent standard deviation (SD)

4EB-P1 in response to insulin stimulation demonstrated similar dynamics between CTRL and siSMP podocytes, while SMP OE podocytes showed no changes in AKT phosphorylation over time, but increased phosphorylation of p70S6K and 4EB-P1 (Fig. 2c). We therefore focused on SMP OE podocytes in subsequent experiments. The modulation of IR signaling by SMPDL3b in podocytes was not linked to a change in IR expression (Fig. 2d). However, subcellular fractionation experiments demonstrated a decreased presence of IR at the PM in SMP OE podocytes compared to control podocytes (Fig. 2e), which could at least partially explain why insulin-stimulated AKT phosphorylation is impaired. Further experiments using flow cytometry assay validated our findings and showed that SMP OE podocytes have less IR-positive cells (Fig. 2f). The displacement of IR from the PM was independent of caveolin-1, a lipid raft-associated protein required for proper insulin signaling[14–16], as phosphorylated and total amounts of caveolin-1 in whole cell lysates (Supplementary Fig. 2a) and in the PM fraction (Supplementary Fig. 2b) remained unchanged. The opposite regulation of AKT and p70S6K phosphorylation after insulin stimulation observed in SMP OE podocytes prompted us to investigate if SMPDL3b may contribute to the functional diversification of IR isoforms signaling, as demonstrated in pancreatic beta cells, where IRA, in which exon 11 is excluded, is a modulator of p70S6K phosphorylation while IRB is a regulator of AKT phosphorylation[13]. More detailed analysis of IR expression revealed that human podocytes express both IR isoforms[17], with IRB being the

dominantly expressed isoform in both SMP OE and control podocytes (Fig. 2g). As tools to experimentally distinguish between endogenous IRA and IRB protein expression are not available, additional experiments in transfected HEK293 cells were performed to determine whether SMPDL3b differentially interferes with IRA/IRB signaling.

**SMPDL3b competes with binding of IRA and IRB to caveolin-1.** Given that we found a differential regulation of the insulin-stimulated PI3K-AKT and mTOR-signaling pathways in SMP OE podocytes, we tested the hypothesis that SMPDL3b over-expression differentially affects IRA and IRB signaling by interfering with the formation of IRA/caveolin-1 or IRB/caveolin-1 complexes. Immunoprecipitation (IP) experiments of exogenous proteins co-expressed in HEK293 were performed. We found that SMPDL3b immunoprecipitates both IR isoforms (Fig. 3a) and caveolin-1, and the ability to form complexes with IR isoforms and caveolin-1 was not affected by the utilization of the SMPDL3b phosphodiesterase mutant SMPDL3b-H135A (Fig. 3b). Considering that SMPDL3b interacts with IRA, IRB, and caveolin-1, we investigated if SMPDL3b competes with IR isoforms to bind caveolin-1 by competitive IP studies in trans-fected HEK293 cells. We found that SMPDL3b facilitates the interaction of caveolin-1 and IRA (Fig. 3c, left panel) while interfering with the ability of IRB to bind caveolin-1 (Fig. 3c, right panel), even when expressed at low levels. To confirm the

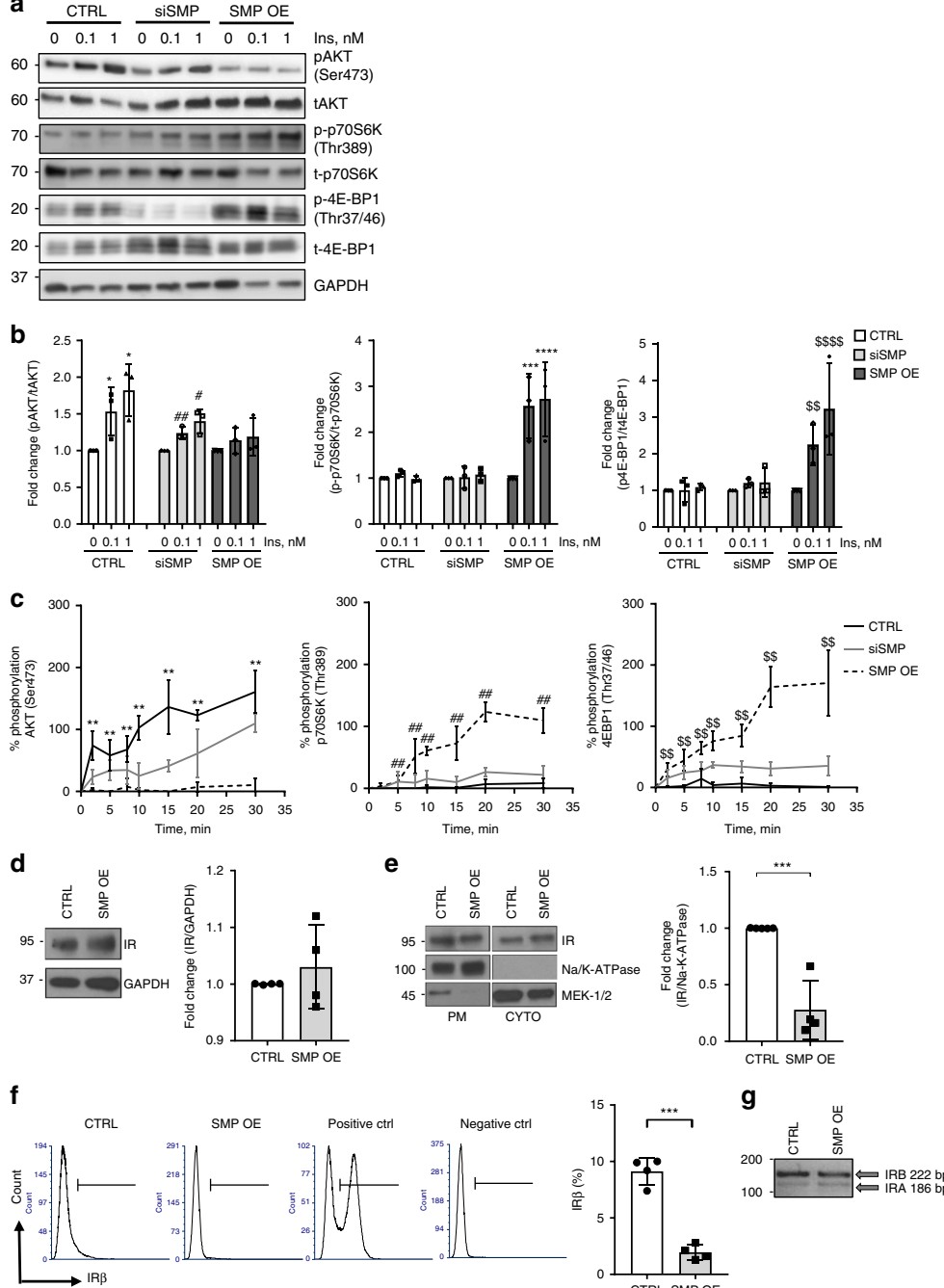

interaction of SMPDL3b with IRA and IRB ex vivo, endogenous IP experiments using pooled mouse glomeruli were performed and showed that SMPDL3b precipitates both IR isoforms and caveolin-1 (Fig. 3d). While in the absence of SMPDL3b, caveolin-1 preferentially binds IRB, the presence of SMPDL3b interferes with the binding of IRB to caveolin-1 and facilitates binding of IRA to caveolin-1, which is consistent with the observation that insulin-stimulated AKT phosphorylation is impaired while p70S6K phosphorylation is augmented in SMP OE podocytes (Fig. 2a, b). As binding of IR to caveolin-1 is important to promote insulin signaling[14,18–20], our data suggest that SMPDL3b overexpression may shift IR signaling from IRB-mediated to IRA-mediated signaling. Next, we tested whether methyl-β-cyclodextrin (CD), which was found to act as a chemical caveolin-1 inhibitor[21], can restore IRB signaling in SMPDL3b-

overexpressing cells. We found that CD increased AKT phosphorylation (Fig. 3e) without affecting p70S6K phosphorylation (Fig. 3e). Additionally, CD treatment of transfected HEK293 cells resulted in restored interaction between IRB and caveolin-1 in presence of SMPDL3b overexpression, while IRA/caveolin-1 interaction was compromised (Fig. 3f). This occurred in association with increased IR and decreased SMPDL3b localization at the PM (Fig. 3g). While it is not possible to discriminate with this model if CD differentially affects IRA or IRB localization, the observed increase in AKT phosphorylation in the absence of changes in p70S6K phosphorylation suggests that CD primarily restores the PM localization of IRB.

**Podocyte-specific *Smpdl3b*-deficient mice have normal kidneys.** The generation of *Smpdl3b* floxed mice is described in detail in

**Fig. 2** SMPDL3b regulates the insulin-signaling pathway in human podocytes. **a** Representative Western blot image of main downstream targets of the insulin receptor signaling in control (CTRL), SMPDL3b knockdown (siSMP) and SMPDL3b overexpressing (SMP OE) podocytes exposed to increasing concentration of insulin (0, 0.1, 1 nM): phosphorylated (pAKT) and total (tAKT) AKT; phosphorylated (p-p70S6K) and total (t-p70S6K) p70S6K; phosphorylated (p-4EB-P1) and total (t-4EB-P1) 4EB-P1. **b** Bar graph analysis of the fold change ratio of pAKT to tAKT (left panel), p-p70S6K to t-p70S6K (central panel) and p-4EB-P1 to t-4EB-P1 (right panel) in CTRL, siSMP, and SMP OE podocytes. $n = 3$ independent experiments; *$P < 0.05$ (when compared AKT phosphorylation level in insulin treated CTRL cells); ##$P = 0.002$, #$P = 0.011$, $F = 5.67$ (when compared AKT phosphorylation level in insulin-treated siSMP cells); ***$P = 0.0002$, ****$P < 0.0001$, $F = 21.47$ (when compared p70S6K phosphorylation level in insulin-treated SMP OE cells); $$$P = 0.008$, $$$ $P < 0.0001$, $F = 10.09$ (when compared 4EB-P1 phosphorylation level in insulin-treated SMP OE cells); one-way ANOVA. $n = 3$ independent experiments. **c** Kinetics of AKT (left panel), p70S6K (central panel), and 4EB-P1 (right panel) phosphorylation in CTRL, siSMP, and SMP OE podocytes treated with 1 nM of insulin. **$P = 0.001$, $F = 9.31$; ##$P = 0.0003$, $F = 12.27$; $$$P = 0.003$, $F = 7.84$, one-way ANOVA. $n = 3$. **d** Western blot and bar graph analysis of insulin receptor (IR) expression in total lysates of CTRL and SMP OE podocytes; $P = 0.448$, two-tailed $t$-test. $n = 4$ independent experiments. **e** Western blot and graph analysis of IR localization at the plasma membrane (PM) in CTRL and SMP OE podocytes. Na/K-ATPase was used as a marker for the PM fraction and MEK-1/2 was used as a marker of the cytosolic fraction (CYTO); ***$P = 0.0004$, two-tailed $t$-test. $n = 5$ independent experiments. **f** FACS and related quantification analysis of the insulin receptor expression in CTRL and SMP OE podocytes. Unstained cells were used as a negative control; HEK293 cells transfected with an insulin receptor overexpressing construct were served as a positive control. A total of 30,000 cells per sample was analyzed; ***$P < 0.0001$, two-tailed $t$-test. $n = 4$ independent experiments. **g** PCR analysis of the insulin receptor isoform A (IRA) and isoform B (IRB) gene expression in CTRL, and SMP OE podocytes. Error bars represent standard deviation (SD)

the "Methods" section. To generate podocyte-specific *Smpdl3b*-deficient mice, *Smpdl3b*-floxed mice were mated with Podocin-Cre mice[22]. A schematic representation of the *Smpdl3b* wildtype allele and targeted allele is shown in Fig. 4a. Genotyping was performed by PCR on genomic DNA isolated from tail biopsies. Sequences of primers used for genotyping are indicated in Supplementary Table 1. Two primer sets were used, one specific for the Podocin-Cre transgene, yielding a 455 bp PCR product and another set to detect either the *Smpdl3b* floxed (237 bp PCR product) or wildtype allele (315 bp PCR product) (Supplementary Fig. 3a). To determine if successful Cre-mediated recombination has occurred, PCR on genomic DNA isolated from glomeruli was also performed using the primers indicated in Supplementary Table 1. This PCR will only yield a product (198 bp) if Cre-mediated recombination has occurred (Supplementary Fig. 3b). However, a 1075 bp band was also detected which could be explained by the presence of mesangial and endothelial cells in glomerular extracts in which Cre-mediated recombination did not occur. Nevertheless, as expected, the 198 bp PCR product was only detected when DNA isolated from glomeruli of heterozygous (*Cre+;fl/+*) and homozygous (*Cre+;fl/fl*) mice was used but not in wildtype mice. We then performed qRT-PCR to determine *Smpdl3b* mRNA expression levels in glomeruli isolated from wildtype and podocyte-specific *Smpdl3b*-deficient mice and demonstrated a 57% reduction of *Smpdl3b* expression in *Cre+;fl/fl* mice (Fig. 4b). As expected, *Smpdl3b* expression levels in tubules remained unchanged (Fig. 4c). Similarly, SMPDL3b protein expression in glomeruli isolated from wildtype (*Cre+;+/+*) and podocyte-specific *Smpdl3b*-deficient mice showed a 55% reduction in *Cre+;fl/fl* mice (Fig. 4d) when compared to wildtype littermates. Finally, SMPDL3b expression levels in podocyte-specific *Smpdl3b*-deficient mice were also found to be reduced in immunofluorescence staining for SMPDL3b. Synaptopodin was used as a podocyte-specific marker (Fig. 4e).

Podocyte-specific *Smpdl3b*-deficient mice are viable and fertile. At the age of 28 weeks *Smpdl3b*-deficient mice demonstrated no changes in body weight (Fig. 4f), kidney to body weight ratio (Fig. 4g), urine albumin to creatinine ratio (Fig. 4h), blood urine nitrogen (Fig. 4i), serum creatinine levels (Fig. 4j), or mesangial expansion score (Fig. 4k) when compared to wildtype mice. Suppression of SMPDL3b expression in podocyte-specific *Smpdl3b* deficient mice was not sufficient to lead to a change in sphingolipid composition in kidney cortexes, suggesting that other cell types may compensate for SMPDL3b deficiency in podocytes (Supplementary Fig. 3c–e; Supplementary Data 4).

**Smpdl3b-deficient mice are protected from DKD.** As podocyte-specific *Smpdl3b*-deficient mice are phenotypically normal, we next tested if podocyte-specific deficiency of *Smpdl3b* is sufficient to protect diabetic mice from DKD. To test this hypothesis, double knockout mice carrying a mutation in leptin receptor (*db/db*) and a podocyte-specific *Smpdl3b* deficiency (*fl/fl*) were generated (Supplementary Fig. 4a). For genotyping, genomic DNA isolated from tail biopsies and PCR primers for genotyping as indicated in Supplementary Table 1 were used.

Four groups of mice were utilized: (1) wildtype control mice (*+/+;+/+*), $n = 10$; (2) podocyte-specific *Smpdl3b* deficient mice (*fl/fl;+/+*), $n = 11$; (3) diabetic control mice (*+/+;db/db*), $n = 9$; and (4) podocyte-specific *Smpdl3b*-deficient diabetic mice (*fl/fl;db/db*), $n = 9$. No changes were found in body weight (Fig. 5a) or kidney to body weight ratio (Fig. 5b) between *+/+;db/db* and *fl/fl;db/db* mice. At 28 weeks of age, *fl/fl;db/db* mice showed significantly lower urinary albumin to creatinine ratios (Fig. 5c) and mesangial expansion (Fig. 5d) compared to *+/+;db/db* mice, suggesting a protective effect of the podocyte-specific deletion of *Smpdl3b* in DKD progression in this model. We did not detect any changes in BUN (two-tail $t$-test; $P = 0.114$), serum creatinine (two-tail $t$-test; $P = 0.466$), glycemia (two-tail $t$-test; $P = 0.067$), or liver function (ALT and AST, two-tail $t$-test; $P = 0.634$ and $P = 0.670$, respectively) between double knockout mice and diabetic mice (Supplementary Table 2), indicating that podocyte-specific *Smpdl3b* deficiency does not lead to any renal or hepatic toxicity. Additionally, no changes were found in total cholesterol or triglyceride levels between *+/+;db/db* and *fl/fl;db/db* mice (Supplementary Table 2). Because our in vitro data suggested an important role of SMPDL3b in the generation of ceramide from C1P, detailed lipidomic analysis of sphingolipids in kidney cortexes of the double knockout mice was performed using LC–MS analysis (Supplementary Data 5). Similar to what we found in SMP OE podocytes (Fig. 1), kidney cortexes from *fl/fl;db/db* mice demonstrated no changes in total ceramide (Fig. 5e) or total sphingomyelin (Fig. 5f) in association with a decreased C1P content (Fig. 5g). Podocyte-specific deletion of *Smpdl3b* in diabetic mice was sufficient to restore normal C1P levels in kidney cortex (Fig. 5g). These results support an important role of SMPDL3b in vivo as an active lipid-modifying enzyme and outline the importance of podocyte SMPDL3b expression in the pathogenesis of DKD.

Additionally, podocyte density and apoptosis were assessed by immunofluorescent labeling using an anti-Wilms' Tumor 1 (WT1) antibody. WT1-positive nuclei were counted in 20 consecutive glomerular cross-sections per animal as reported[11].

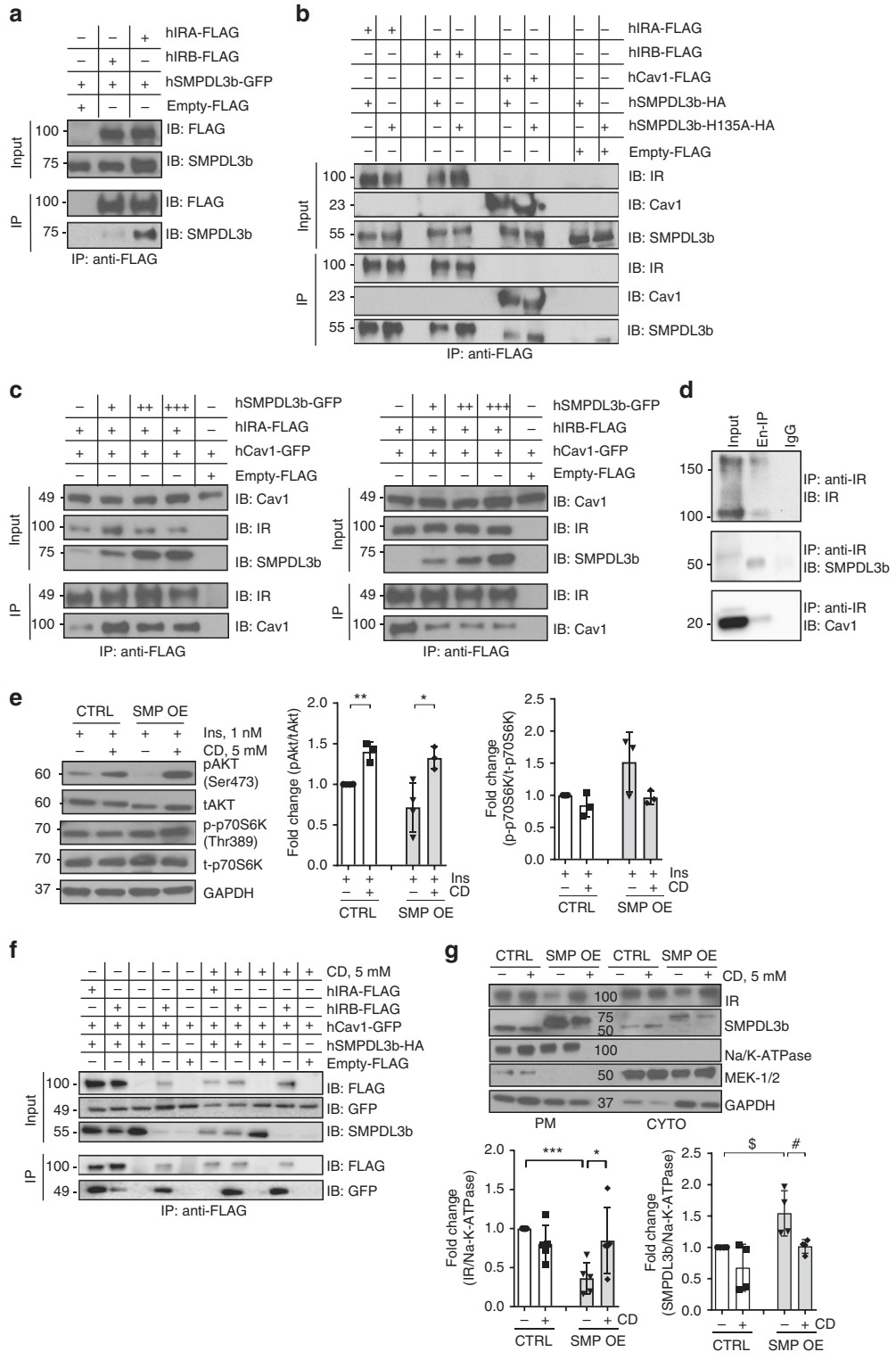

fl/fl;db/db mice demonstrated preserved podocyte number per glomerulus when compared to +/+;db/db mice (Fig. 5h) further supporting a protective effect of decreased podocyte SMPDL3b expression in DKD. Moreover, electron microscopy analysis revealed significant improvement of podocyte foot process effacement and reduced glomerular basement membrane (GBM) thickness in fl/fl;db/db mice compared to +/+;db/db

mice (Fig. 5i) with a tendency to a decreased foot processes width (Supplementary Fig. 4b). Immunohistochemistry analysis showed that fl/fl;db/db glomeruli were characterized by increased phosphorylated AKT compared to +/+;db/db mice (Fig. 5j). These results prompted us to further investigate the link between podocyte-specific SMPDL3b deficiency, C1P availability and insulin signaling in the protection from DKD.

**Fig. 3** SMPDL3b interacts with IRA, IRB, and caveolin-1 in a competitive manner. **a–c** Immunoprecipitation (IP) experiments performed in HEK293 cells co-transfected with human hIRA-FLAG, hIRB-FLAG, hSMPDL3b-GFP, hSMPDL3b-HA hSMPDL3b-H135A-HA, hCav1-FLAG, and hCav1-GFP plasmids. Empty FLAG vector (Empty-FLAG) was used as a negative control. Antibodies (IB) against FLAG, SMPDL3b, insulin receptor β-subunit (IR), or caveolin-1 (Cav1) were used in Western blot analysis. Each IP experiment was repeated at least three times. **a** SMPDL3b interacts with both IR isoforms but preferentially with IRA. **b** Wildtype SMPDL3b interacts with IRA, IRB, and caveolin-1, a mutation (H135A) in the phosphodiesterase activity domain of SMPDL3b does not affect these interactions. **c** Overexpression of SMPDL3b enhances the interaction between caveolin-1 and IRA (left panel) and suppresses the interaction between caveolin-1 and IRB (right panel). **d** Endogenous IP experiments using glomeruli isolated from five C57BL/6 mice indicate that SMPDL3b immunoprecipitates insulin receptor and caveolin-1. IgG served as a negative control. Each IP was repeated three times. **e** Western blot and bar graph analysis of AKT and p70S6K phosphorylation in control (CTRL) and SMPDL3b overexpressing (SMP OE) podocytes pre-treated with 5 mM methyl-β-cyclodextrin (CD) and stimulated with 1 nM insulin; $*P = 0.02$, $**P = 0.001$, two-tailed $t$-test. $n = 3$ independent experiments. **f** IP experiments performed in HEK293 cells co-transfected with human hIRA-FLAG, hIRB-FLAG, hCav1-GFP, and hSMPDL3b-HA show that 5 mM CD treatment abrogates IRA/Cav1 interaction and restores IRB/Cav1 interaction in presence of SMPDL3b overexpression. **g** Western blot and bar graph analysis of the insulin receptor (IR) and SMPDL3b localization at the plasma membrane (PM) in CTRL and SMP OE podocytes exposed to 5 mM CD for 1 h; $*P = 0.049$, $***P = 0.0001$ (for IR); $^\$P = 0.024$, $^\#P = 0.031$ (for SMPDL3b); two-tailed $t$-test. $n = 3$ independent experiments. Error bars represent standard deviation (SD)

**C1P supplementation restores IR signaling and protects from DKD.** Given our observation that podocyte-specific *Smpdl3b*-deficient diabetic mice are protected from experimental DKD, and that kidney cortex from *db/db* mice are characterized by decreased C1P content (Fig. 5g), we hypothesized that exogenous supplementation of C1P may protect from albuminuria and podocyte injury in DKD. We therefore tested if exogenous C1P supplementation is sufficient to restore podocyte insulin signaling in vitro and to protect from DKD in vivo. Control and SMP OE podocytes were pre-treated with 100 μM of synthetic C1P C16:0 for 1 h and then stimulated with 1 nM of human insulin for 30 min. We found that treatment of SMP OE podocytes with C1P is sufficient to restore AKT phosphorylation (Ser473) in response to insulin stimulation (Fig. 6a), while no changes in phosphorylation of p70S6K were observed (Fig. 6b). Mass spectrometry analysis of ceramide species in kidney cortex of *db/db* mice demonstrated that *db/db* mice produce significantly less C16:0 ceramide compared to heterozygous *db/+* littermates (Fig. 6c), similarly to what we showed in SMP OE podocytes (Supplementary Data 1). To determine whether exogenous C1P administration ameliorates features of DKD in these mice, 12-week-old *db/db* mice with established albuminuria were injected intraperitoneally with 30 mg/kg C16:0 C1P ($n = 6$) or 0.9% normal saline ($n = 6$) daily for 28 days. Heterozygous (*db/+*) littermates injected with 0.9% normal saline served as control ($n = 6$). C1P administration to diabetic *db/db* mice with established DKD resulted in significant improvement of albuminuria when compared to saline injected *db/db* control mice (Fig. 6d). Additionally, *db/db* mice treated with C1P demonstrated significantly less mesangial expansion (Fig. 6e). C1P treatment of *db/db* mice did not negatively affect kidney function (BUN and serum creatinine), liver function (serum ALT, AST), or serum lipids (triglyceride and cholesterol) (Supplementary Table 3). Additionally, no changes in body weight (Supplementary Fig. 5a) or glycemia (Supplementary Fig. 5b) were detected. Pathological analysis at the C1P injection site by standard immunohistochemistry did not reveal any detectable microscopic change (Supplementary Fig. 5c). In addition, C1P treatment of *db/db* mice resulted in increased podocyte AKT phosphorylation when compared to *db/db* normal saline-injected mice (Fig. 6f). To investigate if observed improvement of the kidney function with exogenous C1P administration in *db/db* mice is related to the improvement of glycemic control, we performed intraperitoneal glucose tolerance test. No changes were found in fasting glycemia (Fig. 6g, left panel) or intraperitoneal glucose tolerance test (Fig. 6g, right panel) between saline injected and C1P-treated *db/db* mice.

## Discussion

Our study identifies SMPDL3b as a sphingolipid-related enzyme that influences the content of active lipids and acts as a master regulator of insulin signaling in lipid raft domains of the PM. Our findings suggest a model by which SMPDL3b may affect podocyte function and survival, where excessive SMPDL3b expression may cause the displacement of IR from caveolin-1-rich domain in a C1P-dependent manner, resulting in impaired ability to phosphorylate AKT thus promoting podocyte injury in vitro and development of DKD in vivo.

We first confirmed the presence of increased phosphodiesterase activity in SMP OE podocytes when compared to wildtype, which was expected given the homology of SMPDL3b with acid sphingomyelinase and the recently reported crystallographic structure[5]. Unexpectedly, this activity was not associated with a reduction of the sphingomyelin content and increased ceramide production in podocytes, which prompted us to utilize a large-scale lipidomic approach to identify additional sphingolipid species that may be affected by SMPDL3b. Among the different lipid species, a reduction of C1P species was observed, predominantly C16:0 C1P. Interestingly, the analysis of the amino acid sequence of SMPDL3b revealed the possible existence of a phosphatase domain. We therefore investigated if SMPDL3b may be involved in the conversion of synthetic C1P to ceramide. While experiments in transfected HEK293 cells strongly suggested that SMPDL3b may convert C1P into ceramide, utilization of purified soluble recombinant human SMPDL3b did not confirm a direct enzymatic activity of SMPDL3b, consistent with the fact that other GPI-anchored enzymes may have impaired activity when lacking their GPI anchor[23]. These data, in conjunction with our prior report that SMPDL3b overexpression is also associated with increased sphingosine-1-phosphate (S1P) levels[24], suggest that SMPDL3b regulates the content of active sphingolipids, such as S1P and C1P. In nephrology, S1P has been studied extensively in the context of acute kidney injury[25] and has recently generated attention when a mutation of the gene encoding for S1P lyase, resulting in accumulation of S1P, was found to be associated with nephrotic syndrome[9], similar to what was described in S1P lyase-deficient mice[7]. On the contrary, much less is known about C1P, as its pro-inflammatory or anti-inflammatory role is still a matter of debate[6,26–28]. C1P has been shown to modulate AKT phosphorylation in skin fibroblasts, hematopoietic cells[29], macrophages[30], adipocytes[31], consistent with the observation that several sphingolipids are major modulators of insulin signaling[32,33]. As ceramide has been shown to be a master regulator of insulin signaling and the primary antagonist of protein kinase B (AKT) in culturing myotubes and liver[34–36], the fact that SMPDL3b affects overall C1P content without affecting ceramide

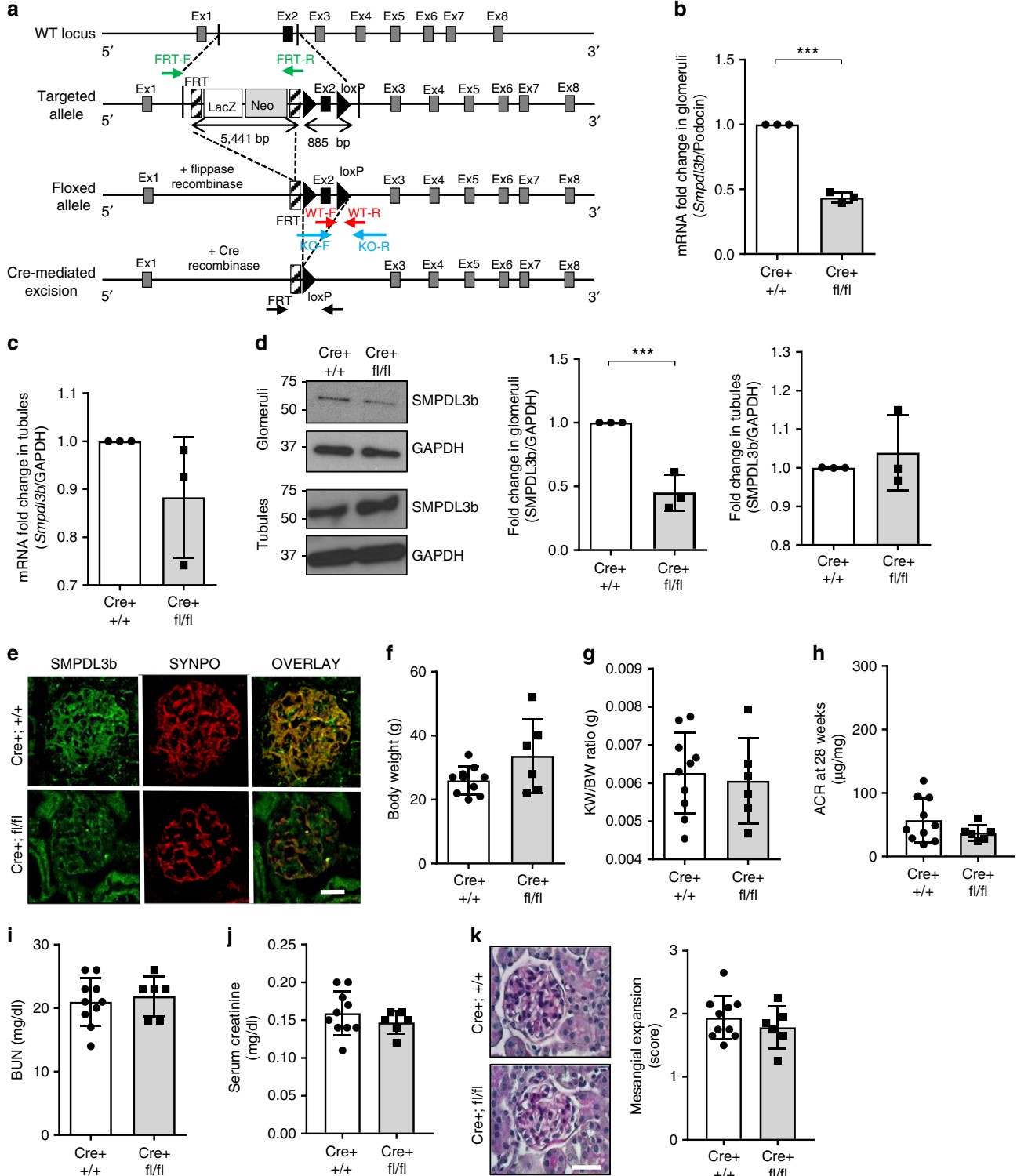

**Fig. 4** Podocyte-specific *Smpdl3b*-deficient mice are viable and phenotypically normal. **a** A schematic representation of the *Smpdl3b* wildtype allele and targeted locus. Ex1–8 exons 1–8; primers to detect FRT (green arrowheads), the *Smpdl3b* floxed allele (KO, blue arrows) and *Smpdl3b* wildtype allele (WT, red arrows) were used. **b, c** RT PCR of *Smpdl3b* expression in glomeruli (**b**) and tubules (**c**) pooled from five different mice; ***$P = 0.003$, two-tailed *t*-test. $n = 3$ independent experiments. **d** Western blot (left panel) and bar graph analysis of SMPDL3b expression in glomeruli (middle panel) and tubules (right panel) pooled from 2 to 5 mice per group; ***$P = 0.003$, two-tailed *t*-test. $n = 3$ independent experiments. **e** Immunostaining (40×) of SMPDL3b (green) and synaptopodin (red; SYNPO) expression in *Cre+;+/+* and *Cre+;fl/fl*. Bar scale 30 μm. **f–j** Phenotypical analysis of *Cre+;+/+* ($n = 10$ animals) and *Cre+; fl/fl* ($n = 6$ animals) 28-week-old mice. *Cre+;+/+* or *Cre+;fl/fl* mice showed no changes in body weight (**f**), $P = 0.076$, kidney weight to body weight ratio (**g**), $P = 0.715$, urine albumin-creatinine ratio (ACR) (**h**), $P = 0.208$, blood urine nitrogen (BUN) (**i**), $P = 0.659$ and serum creatinine(**j**), $P = 0.352$; two-tailed *t*-test. **k** PAS staining (20×) of 4 μm kidney sections and bar graph analysis in *Cre+;+/+* and *Cre+;fl/fl* mice. Bar scale 30 μm. $P = 0.401$, two-tailed *t*-test. Error bars represent standard deviation (SD)

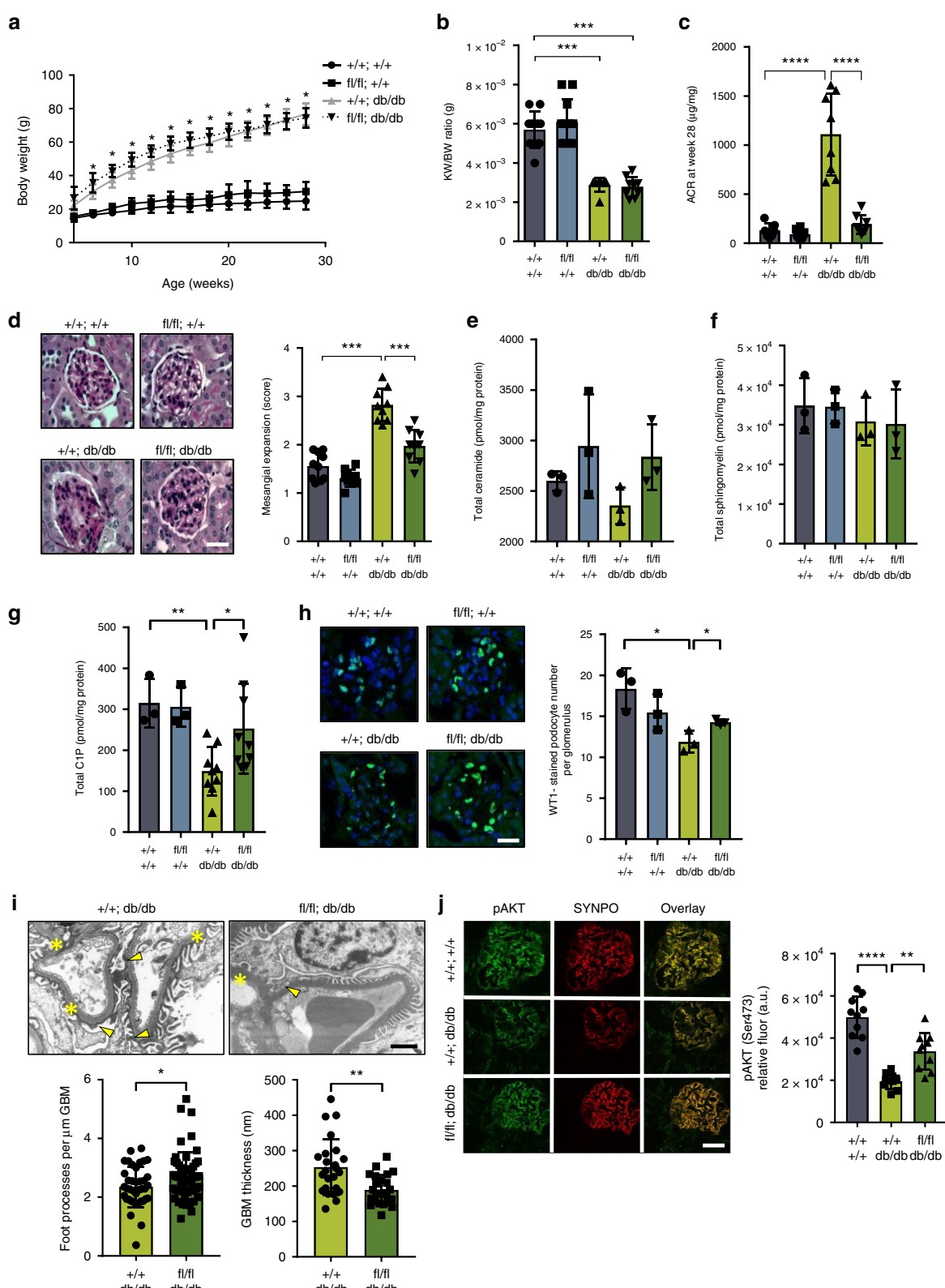

does not exclude the possibility that SMPDL3b influences the ceramide/C1P ratios in PMs of podocytes, thus contributing to insulin resistance. It is possible that the increased ceramide generation in SMPDL3b overexpression cells still plays a key role in the induction of insulin resistance and that our failure to detect a change in ceramide content is linked to its rapid degradation and/or conversion to other substrates. However, should ceramide rather than C1P deficiency be the major driver of insulin resistance in our model, then the administration of C1P in diabetes would result in increased ceramide production and further

**Fig. 5** Podocyte-specific *Smpdl3b*-deficient diabetic mice are protected from DKD. Four groups of 28 weeks-old mice were utilized: (1) wildtype mice (+/+; +/+), n = 10 animals; (2) mice with podocyte-specific *Smpdl3b* deficiency (fl/fl;+/+), n = 11 animals; (3) diabetic mice (+/+;db/db), n = 9 animals; (4) podocyte-specific *Smpdl3b*-deficient diabetic mice (fl/fl;db/db), n = 9 animals. **a** A graph showing body weight changes in all four groups of mice during the study. **b** A bar graph showing kidney to body weight (KW/BW) ratio in all four groups of mice at sacrifice time point. ***$p < 0.0001$, $F = 44.79$, one-way ANOVA. **c** A bar graph showing urine albumin to creatinine ratio (ACR) in all four groups of mice at sacrifice time point. ****$p < 0.0001$, $F = 42.24$, one-way ANOVA. **d** PAS staining (40×) of 4 μm kidney sections and bar graph analysis in all four groups of mice. Bar scale 30 μm. ****$P < 0.001$, $F = 47.17$, one-way ANOVA. **e–g** LC–MS analysis of total ceramide (**e**), total sphingomyelin (**f**) and total ceramide-1-phosphate (**g**) content in kidney cortexes of all four groups of mice. $P = 0.317$, $F = 1.38$ (total ceramide), $P = 0.764$, $F = 0.388$ (total sphingomyelin), one-way ANOVA; *$P = 0.023$, **$P = 0.002$ (total ceramide-1-phosphate), two-tailed $t$-test. n = 3 animals per group. **h** Representative immunostaining (40×) for anti-Wilms' Tumor 1 (WT1, green) and DAPI (blue) in 4 μm kidney sections of all four groups of mice. Bar scale 30 μm. *$P = 0.016$, **$P = 0.037$, two-tailed $t$-test. n = 3 animals per group. **i** Representative TEM and bar graph analysis of foot process effacement (yellow arrow) and glomerular basement membrane (GBM) thickness (yellow asterisk) in +/+;db/db and fl/fl;db/db mice. Bar scale 500 nm. *$P = 0.020$, **$P = 0.001$, two-tailed $t$-test. n = 3 animals per group, n = 10 micrographs analyzed per mouse. **j** Representative immunostaining (40×) for pAKT (Ser473) (green) and synaptopodin (red; SYNPO) in 4 μm kidney sections and bar graph analysis demonstrating increased AKT phosphorylation in all four groups of mice. Bar scale 30 μm. **$P = 0.001$, ****$P < 0.0001$, $F = 37.48$, one-way ANOVA. n = 3 animals per group. Error bars represent standard deviation (SD)

worsening of DKD, and the observed improvement of AKT phosphorylation in vitro and in vivo would remain unexplained. The effect of SMPDL3b on insulin signaling, in association with the observation that SMPDL3b is a lipid raft-associated enzyme[4], prompted us to test the hypothesis that SMPDL3b may interfere with the PM localization and function of IR. We first verified that genes involved in the PI3K-AKT signaling pathway were affected in cells overexpressing SMPDL3b. In SMP OE podocytes, the ability of insulin to cause AKT phosphorylation was severely impaired, while the ability to phosphorylate p70S6K was augmented. These functional data suggested a differential regulation of IR isoform signaling, as it was shown before that in pancreatic beta cells where IRB signaling is primarily linked to AKT phosphorylation and dependent on its localization in lipid raft domains for proper function[13]. Increased SMPDL3b expression was not associated with a change in IR expression but affected IR localization to PM. While it is not possible to discriminate if this change in localization affects IRA or IRB, it seems feasible to suggest that it primarily affects IRB, which is the predominantly expressed isoform in podocytes. We therefore tested if SMPDL3b interferes with the localization of the IR to the PM by interfering with IR binding to caveolin-1. We utilized HEK293-transfected cells and found that SMPDL3b can bind to either IRA or IRB or caveolin-1. Endogenous SMPDL3b binding to IR and caveolin-1 was confirmed by IP in glomeruli isolated from kidney cortexes. Binding of SMPDL3b to caveolin-1 or to IR did not require its phosphodiesterase function, suggesting the existence of an enzymatic regulation of protein complex formation affecting insulin signaling in lipid raft domains. Since the interaction between IR and caveolin-1 has been proven to be essential for IR signaling[37–39], and accumulation of the ganglioside GM3 to caveolae microdomains causes the dissociation of IR-caveolin-1 complexes and increased IR motility in adipocytes[15], we determined if SMPDL3b interferes with the ability of IR to interact with caveolin-1. Indeed, we demonstrated that SMPDL3b can interfere with the binding of IRB to caveolin-1, while facilitating IRA binding to caveolin-1, which may be responsible for increased insulin-stimulated p70S6K phosphorylation. Treatment with methyl-beta-cyclodextrin, which was found to act as a chemical caveolin-1 inhibitor[21], reduced SMPDL3b PM localization and was sufficient to restore IR localization to PM and to restore the ability of insulin to phosphorylate AKT. The ability of sphingolipids to affect insulin signaling has been studied in several organs. Sarcolemmal ceramides and sphingomyelins regulate insulin sensitivity in human skeletal muscles[40]. In adipocytes, S1P deficiency can lead to insulin resistance via decreased activity of the peroxisome proliferator-activator receptor γ[41] or via accumulation of GM3, which has been shown to dissociate the IR beta subunit from caveolin-1[15]. Earlier studies also demonstrated that

high ceramide concentrations are associated with insulin resistance in rodent liver, as well as in skeletal muscles from rats and obese patients[42,43]. Summers et al. (2010) demonstrated that ceramide blocks insulin stimulation of AKT by two independent pathways: by blocking AKT translocation to the PM and by dephosphorylation of AKT by protein phosphatase 2A[44]. To our knowledge, however, this is the first report of how sphingolipids may differentially affect IR isoform signaling by the displacing IRB isoform from caveolin-1 rich domains of the PM.

In order to understand the in vivo relevance of SMPDL3b in the context of diabetic complications that are linked to impaired insulin signaling such as DKD, we generated podocyte-specific *Smpdl3b*-deficient mice. These mice are phenotypically normal, similar to what we described for SMPDL3b-deficient podocytes[2]. Generation of *db/db* mice with a podocyte-specific deletion of *Smpdl3b* resulted in a protection from developing DKD, with reduced albuminuria and mesangial expansion and preservation of podocyte numbers. This occurred in association with the restoration of the C1P content in kidney cortexes, and restoration of podocyte-specific AKT phosphorylation. C1P treatment was not associated with renal or liver toxicity or with changes in the concentration of circulating lipids. In order to understand if the protection observed in *Smpdl3b*-deficient mice was linked to the decreased C1P content, we performed additional in vitro and in vivo experiments with exogenous supplementation of C1P. Similar to what was observed in skin fibroblasts[30], exogenous C1P administration was associated with increased AKT phosphorylation in vitro and with protection from albuminuria and mesangial expansion in *db/db* mice in vivo. These findings open the door for the use of active lipids as potential new therapeutic agents in the field of diabetes and associated complications. In fact, modulation of enzymes involved in sphingolipid metabolism, such as acid and neutral ceramidases has been studied as a therapeutic option in Alzheimer's disease[45,46], cancer research[47,48], and diabetes[41,46]. Interestingly, and similar to what we have observed, an imbalance in C1P and S1P availability is also critical in the progression of other diseases, such as glioblastoma[49], acute lung injury[50,51], Parkinson's disease[52], pancreatic cancer cells[53], and chronic inflammatory processes[54,55]. Nevertheless, little is known about the potential utilization of C1P and S1P analogs as therapeutic agents.

In conclusion, we demonstrated a mechanism by which sphingolipids may affect PM signaling (Fig. 6h). We have utilized IR to demonstrate how active lipid products of SMPDL3b may interfere with IR signaling in lipid raft domains and how administration of deficient lipid species may restore IR signaling and protect from DKD. Our study has limitations, as the technical inability to differentiate endogenous IRA and IRB signaling or to determine the enzymatic activity of recombinant non-GPI-

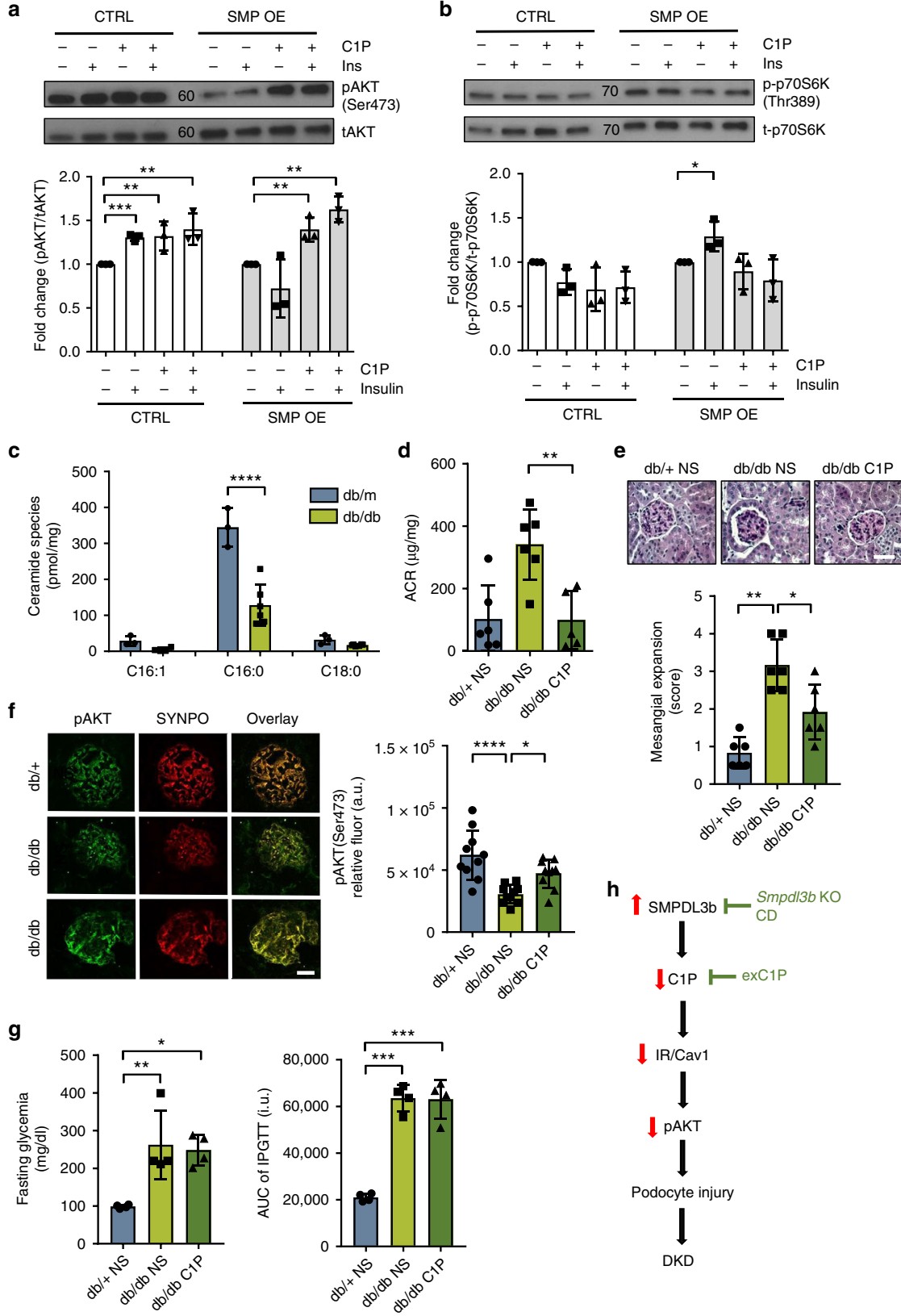

anchored SMPDL3b. In addition, while our studies have been limited to the kidney, understanding the function of SMPDL3b in other organs and in other disease processes may lead to additional therapeutic potential. As pharmacological suppression of

SMPDL3b may render cells susceptible to other forms of injury and an activation of innate immunity pathways[2–4], the alternative approach to utilize C1P replacement strategies may result in a safer and more effective strategy to treat DKD. Further studies

**Fig. 6** Exogenous C1P replacement restores the ability of SMPDL3b overexpressing podocytes to phosphorylate AKT and reduces albuminuria in diabetic mice. **a, b** Western blot and bar graph analysis of phosphorylated AKT (pAKT) to total AKT (tAKT) (**a**) and phosphorylated p70S6 kinase (p-p70S6K) to total p70S6 kinase (t-p70S6K) (**b**) in control (CTRL) and SMPDL3b overexpressing (SMP OE) human podocytes pre-treated with synthetic C1P (100 μM) and treated with insulin (1 nM); *$P < 0.05$, **$P < 0.001$, ***$P < 0.0001$, $F = 9.66$ (for AKT), $F = 3.94$ (for p70S6K), one-way ANOVA. $n = 3$ independent experiments. **c** LC–MS analysis of ceramide species in kidney cortexes of control mice (*db/+*) and diabetic mice (*db/db*). ****$P < 0.0001$, $F = 27.18$, two-way ANOVA. $n = 3$ animals per group. **d** For exogenous C1P administration in vivo three groups of 12-week-old mice were utilized: (1) control heterozygous mice intraperitoneally (I.P.) injected with 0.9% normal saline (*db/+NS*), (2) diabetic mice I.P. injected with 0.9% normal saline (*db/db NS*), 3) diabetic mice I.P. injected with 30 mg/kg C1P for 28 days (*db/db C1P*). $n = 6$ animals per group. A bar graph showing albumin–creatinine ratio (ACR) in *db/+NS*, *db/db NS* and *db/db C1P* mice. **$P = 0.002$, $F = 10.1$, one-way ANOVA. **e** Representative PAS staining (20×) in 4 μm kidney sections and bar graph analysis in *db/+NS*, *db/db NS*, and *db/db C1P* mice. Bar scale 20 μm. *$P = 0.043$, **$P = 0.002$, $F = 9.47$, one-way ANOVA. **f** Representative immunostaining (40×) for pAKT (Ser473) (green) and synaptopodin (red; SYNPO) in 4 μm kidney sections and bar graph analysis in *db/+NS*, *db/db NS*, and *db/db C1P* mice. Bar scale 30 μm. *$P = 0.044$, ****$P = 0.0001$, $F = 8.03$, one-way ANOVA. **g** Bar graph analysis of fasting glycemia (left panel) and area under curve of glucose tolerance test (right panel) after intraperitoneal injection of 1.5 g/kg D-glucose in *db/+NS*, *db/db NS*, and *db/db C1P* mice. *$P = 0.013$, **$P = 0.008$, ***$P < 0.0001$, $F = 9.86$ (for fasting glycemia), $F = 67.31$ (for AUC of IPGTT), one-way ANOVA. Error bars represent standard deviation (SD). **h** Proposed mechanism by which increased SMPDL3b expression contributes to the pathogenesis of DKD and possible targeting strategies to prevent DKD development. (↑) overexpression, (↓) suppression, CD - cyclodextrin, exC1P - exogenous C1P

investigating C1P administration in vivo will open the door to therapies with bioactive lipids targeting the most prevalent form of kidney disease worldwide.

## Methods

**Reagents, antibodies, dilution, and mice.** RPMI 1640 media for podocytes culture was obtained from Corning (NY, USA; #10-040-CV). DMEM media with 4.5 g/l glucose, L-glutamine and sodium bicarbonate (#11995-073) for HEK293 cell culture was obtained from ThermoFisher Scientific Inc. (CA, USA). Dynabeads M-270 Epoxy Co-IP Kit (#14321D) was obtained from ThermoFisher Scientific Inc. Human insulin (#I9278-5ML), CD (#H107-100G), 4-nitrophenol standards (#1048-5G) and bis-*p*-nitrophenolphosphate (#N3002) for phosphodiesterase assay were obtained from Sigma-Aldrich (MO, USA). C16:0 C1P (#860533) and C6-NBD-Ceramide (#810209P) were obtained from Avanti Polar Lipid (AL, USA). Recombinant human SMPDL3b protein, MYC/DDK-tagged was ordered from Creative Biomart (#SMPDL3b-2823H).

Anti-caveolin-1 (#3251 for phospho-Cav1(Tyr14), dilution 1:1000, #3267 for Cav1 (D46G3) XP™, dilution 1:1000), anti-IR β subunit (4B8) (#3025, dilution 1:100), anti-Na/K-ATPase (#3010, dilution 1:1000), anti-MEK-1/2 (D1A5) (#8727, dilution 1:1,000), anti-AKT (#9271 for phospho-AKT (Ser473) (D9E), dilution 1:1000, #9272 for tAKT, dilution 1:1000), anti-p70S6Ke (#9234 for phosphor-p70S6K (Thr389 (108D2)), dilution 1:1000, #9202 for p70S6K, dilution 1:1000), anti-4EBP1 (#2855 for phospho-4E-BP1 (Thr37/46) (236B4), dilution 1:1000, #9644 for 4E-BP1, dilution 1:1000) primary antibodies were purchased from Cell Signaling Technology (MA, USA). SMPDL3b antibodies purchased from GenWay Biotech (CA, USA, #GWB-2281D4, dilution 1:1000). IR β subunit antibodies (CT-1) for endogenous IP (#MA5-13778, dilution 1:250) were obtained from ThermoFisher Scientific Inc., HRP-conjugated secondary antibodies from Promega Corp. (WI, USA) and used in dilution 1:10,000, anti-rabbit TrueBlot HRP-conjugated secondary antibodies from Rockland (PA, USA, #18-8816-31, dilution 1:2,000). For immunofluorescence primary rabbit polyclonal anti-WT1 (WT1; #sc-192, dilution 1:300) and anti-goat polyclonal synaptopodin (P19; #sc-21537, dilution 1:500) antibodies were purchased from Santa Cruz Biotechnology (TX, USA), rabbit polyclonal SMPDL3b antibodies from GenWay (#GWB-2281D4, dilution 1:200) and rabbit monoclonal pAKT (Ser473; #9271, dilution 1:300) antibodies from Cell Signaling Technology. Alexa fluorescence 568 (red; #A10042; #11061, dilution 1:500), 488 (green; # A11059, dilution 1:500) secondary antibodies and DAPI (#D1306) were obtained from Invitrogen | Thermo Fisher Scientific (CA, USA). For flow cytometry, human BD Fc Block reagent (#564220) was obtained from BD Bioscience, Human/Mouse Insulin R/CD220 APC-conjugated antibodies (#FAB1544A, dilution 1:10) and isotope control IgG APC-conjugated antibodies (#IC108A, dilution 1:30) were obtained from R&D Systems.

Mice in which exon 2 of *Smpdl3b* is flanked by loxP sites were purchased from the International Knockout Mouse Consortium (B6N;B6N-*SMPDL3b*tm1a (EUCOMM)Wtsi/H; #MGI:1916022). Mice carrying a Flp-recombinase transgene (B6.129S4-*Gt(ROSA)26Sor*tm1(FLP1)Dym/RainJ; #009086), mice carrying a Cre-recombinase transgene specifically expressed in podocytes (B6.Cg-Tg(NPHS2-cre) 295Lbh/J; catalog #008205) and *Lepr*db heterozygous db/+ and homozygous db/db mice (B6.BKS(D)-Lepr<db>/J; catalog #000697) were purchased from the Jackson Laboratories (USA). 18% protein rodent chow diet was ordered from Tekland Global (Envigo, WI, USA; #2018S).

**Plasmids.** A full-length human SMPDL3b with GFP tag (hSMPDL3b-pCMV6-AC-GFP; #RG217688), human caveolin-1 with GFP tag (hCav1-pCMV6-AC-GFP; #RG210274), human caveolin-1 with FLAG tag (hCav1-pCMV6-Myc-DDK; #RC210274), and pCMV6-AC-3DDK empty vector (#PS100057) were ordered from OriGene (MD, USA). A full-length human IR isoform A with FLAG tag (hIRA-

FLAG-pRC/CMV) and a full-length human IR B with FLAG tag (hIRB-FLAG-pRC/CMV) plasmids were a kind gift from our collaborator Dr. Ingo Leibiger[13]. A full-length human HA-tagged SMPDL3b plasmid (pTO-SP-HA-hSMPDL3b) and HA-tagged SMPDL3b plasmid with mutation in 135 histidine (pTO-SP-HA-hSMPDL3b-H135A) were a kind gift from our collaborator Dr. Leonhard Heinz[4].

**Cell culture.** A human podocyte cell lines transfected with a thermosensitive SV40-T construct were cultured on type I collagen-coated flasks in RPMI media supplemented with 10% FBS, 1% penicillin/streptomycin, and 1% ITS at 33 °C to promote cells proliferation[56]. Before thermoswitching to 37 °C to promote cells differentiation, cells were grown to confluence 70–80% and trypsinized and reseeded in fresh flasks at a dilution 200,000 cells per 10 ml of RPMI media without ITS. The origin of normal human podocytes (control) is a kind gift of Dr. Jochen Reiser, Rush University, Chicago. SMPDL3b overexpression human podocytes were developed by Dr. Fornoni and validated in ref. [2].

HEK293 cells were obtained from ATCC (Manassas, VA; #CRL-1573™). HEK293 cells were cultured in DMEM media supplemented with 10% FBS and 1% penicillin/streptomycin at 37 °C. Prior to transfection procedure, cells were grown to confluence 70–80%.

**Treatment of human podocytes.** For treatments, human podocytes were differentiated for 14 days and serum starved for 24 h starting day 13. Stimulation with human insulin was performed at concentrations of 0.1 and 1 nM for 30 min at 37 °C. For kinetics of the insulin signaling over time, podocytes were treated with 1 nM of insulin for 0, 2, 5, 10, 15, 20, 25, and 30 min. Treatment with CD was performed at a concentration of 5 mM for 1 h at 37 °C. C1P was used for treatment at a concentration of 100 μm for 1 h at 37 °C. C1P stock solution (final concentration 1.47 mM) was prepared by sonication of C1P in sterile nanopure water on ice using a probe sonicator, until a clear dispersion was obtained[57]. Untreated wildtype podocytes served as a control.

**LC–MS analysis.** Cell lysates from differentiated control (CTRL) and SMP OE podocytes or kidney cortexes from mice were used for lipidomics analysis. LC–MS analysis was performed in Medical University of South Carolina, Lipidomics shared resource (SC, USA). Cells pellets containing at least $1 × 10^6$ cells per sample in vitro or at least 25 mg of kidney cortex per sample in vivo were subjected to liquid extraction[58].

**Phosphodiesterase enzymatic activity assay.** Enzymatic activity in CTRL and SMP OE podocytes was measured according to a previously published protocol[4]. Briefly, CTRL and SMP OE podocytes were differentiated for 14 days at 37 °C in 96-well plates and starved for 24 h. To measure the phosphodiesterase activity, podocytes were incubated with isotonic Tris-buffered saline in the presence of 1 mM substrate at 37 °C and 5% $CO_2$ for 1 h. Phosphodiesterase activity in WT and SMP OE podocytes was measured by generation of *p*-nitrophenol from bis-*p*-nitrophenolphosphate as absorbance at 405 nm in 100 μl reaction volume.

**C1P phosphatase assay.** C1P phosphatase was assayed in vitro using homogenates from HEK293 cells transfected with hSMPDL3b-pCMV6-Myc-DDK or pCMV6-AC-3DDK essentially as described[59]. Briefly, 50 μM C6-NBD-C1P complexed to 10 μM BSA was incubated with homogenates (2 μg of protein) in 200 mM NaCl, 2 mM EDTA, 100 mM HEPES, pH 7.4 and a protease inhibitor cocktail (1:200) for 10 min at 37 °C. The reaction was terminated by addition of 3.75 volumes of chloroform:methanol (1:2), 1.25 volume of chloroform and 1.25 volumes of 2 M KCl/0.2 M $H_3PO_4$. The lower phase was dried under $N_2$ and lipids subsequently resolved by TLC on Silica Gel plates using chloroform:methanol:

acetic acid: 15 mM CaCl$_2$ (60:35:2:4, v/v/v/v) as developing solvent. For experiments with purified SMPDL3b, 50 μM C6-NBD-C1P complexed to 10 μM BSA was incubated with SMPDL3b in concentrations 0, 100, 500, and 1000 ng following the steps described above.

**Illumina sequencing RNA data analysis.** Preparation and sequencing of RNA libraries was carried out in the John P. Hussman Institute for Human Genomics Center for Genome Technology. Briefly, total RNA was quantified and qualified using the Agilent Bioanalyzer to have an RNA integrity score (RIN) above 7. For each of the six samples, 500 ng of total RNA was used as input for the Illumina TruSeq Stranded Total RNA Library Prep Kit with Ribo-Zero to create ribosomal RNA-depleted sequencing libraries. Each sample had a unique barcode to allow for multiplexing and >35 million reads in a 2 × 100 paired end sequencing run on the Illumina HiSeq3000 were obtained.

Raw sequence data were processed by the on-instrument Real Time Analysis software (v.2.7.7) to basecall files. These were converted to de-multiplexed FASTQ files with the Illumina supplied scripts in the BCL2FASTQ software (v2.17). The quality of the reads was determined with FASTQC software for per base sequence quality, duplication rates, and overrepresented k-mers. Illumina adapters were trimmed from the ends of the reads using Trim Galore! package. Reads were aligned to the human reference genome (hg19) with the STAR aligner (v2.5.0a)[60]. Gene count quantification for total RNA was performed using the GeneCounts function within STAR against the GENCODE v19 transcript file.

Gene count data were input into edgeR software[61] for differential expression analysis. Briefly, gene counts were normalized against total aligned reads for each sample to generate counts per million (cpm) expression value for each gene in each sample. Given the relatively small sample size per group for comparison ($n = 3$ in each group) the exact test implemented in edgeR was used to determine differential expression including a false discovery rate $P$-value.

To discover enriched functional-related gene groups, DAVID bioinformatics resource 6.8 was used.

**Isolation of PMs.** Cells were collected in homogenization media (15 mM KCl, 1.5 mM MgCl$_2$, 10 mM HEPES, 1 mM DTT) and incubated on ice for 5 min. Cell pellets were homogenized five times using insulin syringes and incubated with 2.5 M sucrose solution (250 nM final concentration). Cell pellets were submitted to several centrifugation steps: to separate nuclear fraction (1000 × g, 5 min, 4 °C); to separate mitochondrial and endoplasmic reticulum fraction (10,000 × g, 15 min, 4 °C); to separate PM fraction and cytosolic fraction (100,000 × g, 1 h, 8 °C). Resulting membrane pellets were collected and resuspended in 50 μl of homogenization media followed by Western blot analysis. Cytosolic fractions were concentrated in 3–4 times using 3K concentrators, 20,000 × g, 30 min, 4 °C followed by Western blot analysis. Na/K-ATPase was used as a marker of PM fraction, and MEK-1/2 was used as a marker of the cytosolic fraction.

**Flow cytometry.** CTRL or SMP OE human podocytes were collected at day 14 of differentiation and incubated with 0.5% BSA, 0.1% Saponin in 1 × PBS for 10 min at room temperature followed by Fc blocking step for 10 min at room temperature using 12.5 μg of human BD Fc Block solution. Incubation with 30 μl of IR APC-conjugated antibodies per sample was performed for 1 h at room temperature in dark. HEK293 IR-transfected cells were served as positive control. CTRL human podocytes incubated with 10 μl of isotop control IgG APC-conjugated antibodies were used as a negative control. Fluid sheath running through the cytometer was highly filtered (0.22 μm filters). FACS experiments were performed in four biological replicates and 30,000 events per sample were analyzed. Fluorescence exclusion was used to gate on live cells. Intact cells were identified by conventional flow cytometry using side scatter area (SSC-A) and forward scatter area (FSC-A). IR-positive cells were determined using negative and positive controls gating (Supplementary Fig. 6). Results were read on BD LSR II 15-color flow cytometric analyzer using PE fluorochrome with laser lens 535 nm and emission filters 585/15. All data were analyzed using FACSDiva 8.0.2 software.

**PCR analysis to differentiate IR isoforms.** mRNA from human podocytes was extracted using RNAeasy kit according to the manufacture's protocol. 250 ng RNA was reverse transcribed to obtain cDNA using the high-capacity cDNA reverse transcriptase kit according to the manufacturer's protocol. PCR was performed using GoTaq green master mix at 58 °C, 35 cycles yielding a 222-bp PCR products corresponding to IR isoform B and a 186-bp PCR product corresponding to IR isoform A, using primer sequences reported in Supplementary Table 1[17].

**Transient transfection and IP.** HEK293 cells were grown until 70–80% confluence prior any transfection. Transient transfection was performed using FuGene-6 reagent, OPTI-MEM media, and 1 μg of plasmid and incubated in antibiotic-free DMEM media for 24 h. Then cells were grown in DMEM media supplemented with 10% FBS and 1% penicillin/streptomycin for next 24 h and starved for 2–4 h in DMEM with 0.1% FBS and 1% penicillin/streptomycin prior treatment or cells collection. Cell pellets were collected in Triton buffer (50 mM Tris, pH 7.5, 150 mM NaCl, 1% Triton) and were precipitated with anti-FLAG M2 Affinity Gel[62]. After a wash pellets were analyzed by standard SDS gel

electrophoresis. Endogenous Co-IP was performed on isolated glomeruli from wild-type B6N mice using Dynabeads M-270 Epoxy Co-IP Kit according to manufacturer's protocol. Briefly, Dynabeads were coupled with IR antibodies. Proteins from isolated glomeruli were obtained using extraction buffer (100 mM NaCl, 2 mM MgCl$_2$, 1 mM DTT, 1:200 protease inhibitor without EDTA). Antibody-coupled beads were incubated with lysates (1.5 mg of antibody-coupled beads per 1.5 g of isolated glomeruli) at 4 °C for 30 min, followed by washing steps with buffers (5 min in 200 μl 1 × LWB and 5 min in 60 μl 1xEB) provided by the kit.

**Animal studies.** All animal studies have complied with all relevant ethical regulations and were performed in accordance with the National Institutes of Health Guidelines. The study protocol was approved by the Institutional Animal Care and Use Committee of the University of Miami, Miller School of Medicine.

**Genotyping, housing, and phenotypic analysis of mice.** To generate podocyte-specific Smpdl3b-deficient mice, Cre-loxP technology was used. Mice with a Smpdl3b-targeted allele were crossed to Flp-recombinase carrying mice[63] to generate heterozygous Smpdl3b floxed mice with a deleted FRT-Neo cassette (Fig. 4a, Supplementary Fig. 3). These were then mated with Podocin-Cre Tg/Tg mice[22] to generate heterozygous Podocin-Cre positive Smpdl3b floxed mice (Cre+;fl/+). Finally, the latter were intercrossed to generate homozygous podocyte-specific Smpdl3b-deficient mice (Cre+;fl/fl). As Podocin-Cre and Flp-recombinase transgenic mice were used in a pure C57BL/6 background and floxed Smpdl3b mice were used in a C57BL/6N background, wildtype littermates were used as a control to account for phenotypical variability due to background differences.

For genotyping, tissues from tail biopsies of 3-week-old mice were digested with proteinase K and DNA was isolated using QIAamp DNA Mini kit according to the manufacture's protocol. Smpdl3b, Flp- and Cre- recombinase alleles were detected using PCR (information on primers used for genotyping is available in Supplementary Table 4). Podocin-Cre transgenic mice[22] were identified using primers specific for this transgene, yielding a 455 bp PCR product (Supplementary Fig. 3a). To distinguish wildtype mice from podocyte-specific Smpdl3b-deficient mice, which are characterized by the presence of a 3'-loxP site downstream of exon 2 of the Smpdl3b gene, a genotyping PCR on tail biopsies was established. In this PCR, a set of four primers to detect the wildtype (WT1 and WT2) and the floxed allele (KO1 and KO2) giving rise to a 315 bp and 237 bp PCR product, respectively, was used (Supplementary Fig. 3a). To ensure successful Cre-mediated recombination, which is only detectable in podocytes, PCR on genomic DNA isolated from glomeruli using the primers 5FRT Fw, Smp_Del Rev and FRT_Del_Fw (Supplementary Table 1) was also performed. This PCR will only yield a 198 bp product if Cre-mediated recombination has occurred (Supplementary Fig. 3b). A 1075 bp band can be detected in all other cells of a glomerulus.

Starting at 4 weeks of age, weight measurements and morning spot urines were collected bi-weekly until mice reached 28 weeks of age. Male and female mice were analyzed in all study groups. Six to ten mice per group (Cre+;+/+– wildtype mice ($n = 10$) and Cre+/fl/fl –podocyte-specific Smpdl3b knockout mice ($n = 6$)) were sacrificed and in-depth phenotypical analysis was performed, including urinary albumin-to-creatinine ratio, histological analysis (PAS, H&E), glomeruli isolation and blood analysis.

Heterozygous Podocin-Cre positive Smpdl3b-floxed mice (Cre+;fl/+) were crossed to Lepr$^{db}$ heterozygous db/+ mice to generate double heterozygous mice (Cre+;fl/+;db/+). The latter were intercrossed to generate double mutant mice (Cre+;fl/fl;db/db).

For genotyping, tissues from tail biopsies was digested as described above. Smpdl3b, db, and Cre- recombinase alleles were characterized using PCR (information on primers used for genotyping is available in Supplementary Table 4).

Starting at 4 weeks of age, weight and glycemia level measurements and morning spot urines were collected bi-weekly until mice reached 28 weeks of age. Ten mice per group were sacrificed and in-depth phenotypical analysis was performed, including urinary albumin-to-creatinine ratio, histological analysis (PAS, H&E, PSR), transmission electron microscopy analysis (GBM thickness and foot process effacement quantification), immunohistochemistry, glomeruli isolation, and serum analysis.

All mice were housed in MicroVENT IVC standard ventilated cages (Allentown Inc., PA, USA) maintained at 68 ± 4°F with 12 h dark and 12 h light cycle and provided water and Tekland Global 18% protein rodent chow diet with no restrictions.

**Urine samples analysis.** Morning urine spot was collected bi-weekly. The urine albumin content was measured by ELISA and urinary creatinine was measured by an assay based on the Jaffe method, using albumin ELISA kit from Bethyl Lab. (#E90-134; TX, USA) and creatinine kit from Stanbio (#0420-500; TX, USA). Values are expressed as microgram albumin per milligram creatinine.

**Histology and assessment of mesangial expansion.** After perfusion of an animal with 1 × PBS, the right kidney was removed for histological analysis and the left kidney was collected for glomeruli extraction. Periodic acid-Schiff (PAS) and hematoxylin–eosin (H&E) staining of paraffin-embedded kidney sections (4 μm thick) was performed using a standard protocol. Histological images were

visualized using a light microscope (Olympus BX 41, Tokyo, Japan) at ×40 magnification and analyzed using Image J software[64]. Twenty glomeruli per section were analyzed for mesangial expansion by semi-quantitative analysis (scale 0–4) performed by two blinded independent investigators[65,66].

**Transmission electron microscopy (TEM)**. For TEM, a kidney pole was fixed in 2% paraformaldehyde (PFA), 2% glutaraldehyde in 0.1 M phosphate buffer (pH = 7.4) for at least one week prior to embedding. TEM was performed as described previously[67] with minor modifications. Samples were examined with a JEM-1011 transmission electron microscope (JEOL) in the University of Tokyo. Negatives of electron micrographs at ×10,000 magnification were scanned at 600 d.p.i. Measurements of the resulting images was performed using Image J software. Eight images per mouse were analyzed using the distance between two points on a ruler of a photograph as a measurement scale. GBM thickness was measured in 30 different points. The number of podocyte foot processes (FP) along the GBM was counted by hand. A FP was defined as any connected epithelial segment butting on the basement membrane, between two neighboring filtration pores or slits. From each photograph, the arithmetic mean of the foot process width (FPW) was calculated as follows: $FPW = \pi/4 * \Sigma GBM\ length/\Sigma foot\ process$[68].

**Blood samples analysis**. Blood samples were analyzed for lipid panel, aspartate aminotransferase (AST), alanine transaminase (ALT), and blood urea nitrogen (BUN) in the Comparative Laboratory Core Facility of the University of Miami. Serum creatinine was determined by tandem mass spectrometry at the UAB-UCSD O'Brein Core Center (University of Alabama, Birmingham) as previously described[69].

**Immunofluorescence staining and confocal microscopy**. For immunofluorescence staining of kidneys from mice, fresh cut 4 μm-thick tissue in OCT was used. No fixation agent was applied. Permeabilization was performed using 0.3% Triton X-100 in 1 × PBS for 10 min at room temperature. A blocking step was performed using 5% BSA, 2.5% FBS in PBS for 1 h at room temperature. Primary antibodies were used for 24 h at room temperature. Alexa-conjugated secondary antibodies were used for 1 h at room temperature. To detect nucleus, DAPI staining was applied in a 1:500 dilution for 20 min at room temperature.

To assess podocyte number and apoptosis in murine glomeruli, immunofluorescent labeling with anti-WT1 was performed. WT1-positive nuclei were counted in 20 consecutive glomerular cross-sections per animal by two blinded investigators[11].

Images were acquired by laser scanning confocal microscopy using a Leica SP5 Inverted microscope, ×40 wet objective (Leica Microsystems CMS GmbH, Germany). Measuring of cell fluorescence was performed using Image J software[70].

**Exogenous C1P treatment of diabetic mice**. Six 12-week-old homozygous *db/db* or heterozygous *db/+* mice received daily intraperitoneal injections of 30 mg/kg C1P C16:0 or 0.9% normal saline (NS) for 28 days. To prepare a stock solution (5 mg/ml), C1P C16:0 powder was weighted and dissolved in 0.9% NS and 2.5% DMSO by sonication. Aliquots were stored at −20 °C, then melted and sonicated once again before application. Urine collection, body weight, and glycemia measurements were determined weekly. Mice were sacrificed and in-depth phenotypical analysis was performed, including urinary albumin-to-creatinine ratio, histological analysis (PAS, H&E, PSR), glomeruli isolation and blood analysis as described previously. aspartate aminotransferase (IPGTT) was performed 3 weeks after exogenous C1P administration. After 12 h fasting, blood glucose was measured at baseline, and then at 15, 30, 60, 90, and 120 min after a glucose bolus (1.5 g/kg).

**Statistical analysis and study design**. Data are expressed as a mean ± standard deviation (SD). A number of experiments ranging between 3 and 5 was utilized as indicated for each distinct experiment. Minimal group sizes for in vitro and in vivo studies were determined via power calculator using the DSS Researcher's Toolkit with an α of 0.05. Animals were grouped unblinded, but randomized, and investigators were blinded for the quantification experiments. GraphPad Prism Outlier calculator software was used to indicate outliers in each set of data obtained for in vitro and in vivo experiments. Significant outliers were excluded from further statistical analysis. Two groups of data were compared using the two-tailed *t*-test. Three and more groups of data were compared using the one-way analysis of variance (ANOVA), followed by Tukey's post-test or two-ways ANOVA with multiple comparisons. $P < 0.05$ was taken to indicate statistical significance. Statistical analysis was performed using the GraphPad Prism, version 7.0 (GraphPad Software Inc.).

Uncropped Western blots are depicted in Supplementary Figs. 7–10.

**Reporting summary**. Further information on research design is available in the Nature Research Reporting Summary linked to this article.

## Data availability

The raw data of lipidomic analysis in human podocytes and kidney cortexes of podocyte-specific *Smpdl3b* deficient mice and type 2 diabetic podocyte-specific *Smpdl3b* deficient mice are available in the Supplementary Datas 1, 3 and 5, respectively. All sequencing data supporting the findings of this study are available in the Supplementary Datas 2 and 4 and have been deposited in the National Center for Biotechnology Information Gene Expression Omnibus (GEO) and are accessible through the GEO Series accession number GSE129666. Next publically available sources were used in the study: SMPDL3b predicted domains ("SMPDL3b"). All other relevant data are available from the corresponding author upon reasonable request.

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

## Acknowledgements

A.F. and S.M. are supported by the NIH grants DK104753, DK11759 and CA227493. A.F. is supported by the NIH grants DK090316, U24DK076169, U54DK083912, UM1DK100846, and 1UL1TR000460. A.F., S.M., C.F. are supported by Hoffman-La Roche. G.M.D. is supported by a Predoctoral Fellowship of the American Heart Association (16PRE30200010). JJK is supported by NRSA Post-doctoral fellowship from NIH/NIDDK (F32DK115109). We thank Dr. Chalfant for his suggestions in the design of exogenous C1P experiments in vivo. We thank Dr. Heinz the CeMM Research Center for Molecular Medicine of the Austrian Academy of Sciences for a kind gift of HA-tagged human wildtype and mutant SMPDL3b constructs. We would like to acknowledge the skilled assistance of the Flow Cytometry Shared Resource and the Analytical Imaging Shared Resource of the Sylvester Comprehensive Cancer Center at the University of Miami, Miller School of Medicine, for the provision of the sophisticated fluorescence analysis.

## Author contributions

A.M. performed the in vitro and in vivo experiments, analyzed the data, and wrote methods and results of the manuscript. S.K.M. prepared material for illumina sequencing RNA data analysis, analyzed presence of enriched functional-related gene groups, performed endogenous co-IP experiments, and performed some of the in vivo experiments. G.M.D. performed WT1 histological staining and some in vitro experiments. J.M. performed PSR histological staining and analysis and some in vitro experiments. T.H.Y. initialized the characterization of the podocyte-specific *Smpdl3b* knockout mice colony. E.R-G. and I.D.Z. performed C6 ceramide production assay in HEK293 cells and in podocytes using purified SMPDL3b protein. J.V.S., C.G.H., C.P., M.G., A.S., J.J.K. performed some of the in vitro experiments. J.B. and I.V. performed genotyping procedures and some of the in vitro experiments. C.F. designed some of the immunoprecipitation experiments. Y.Z. and A.J.M. designed experiments related to sphingolipid analysis. Y.Z. analyzed some of the mass spec data. I.L. designed and provided human insulin receptor isoforms plasmids. G.W.B. conceived one of the techniques utilized. A.H.F. designed C6 ceramide production experiments. L.B. evaluated kidney tissue slides for histopathology. Y.I. and R.I. performed TEM experiments. S.M. and A.F. conceived the project, designed

and supervised the study, analyzed the data, and edited the manuscript. A.F. wrote introduction and discussion of the manuscript, she is the guarantor of this work and, as such, had full access to all the data in the study and takes responsibility for the integrity of the data and the accuracy of the data analysis.

## Additional information

**Competing interests:** G.W.B., A.F., and S.M. are inventors on pending or issued patents (US10,183,038, US10,052,345) aimed to diagnose or treat proteinuric renal diseases. They stand to gain royalties from their future commercialization. A.F. is Chief Scientific Officer of L&F Health LLC and is consultant for Variant Pharmaceutical. Variant Pharmaceuticals, Inc. has licensed worldwide rights to develop and commercialize hydroxypropyl-beta-cyclodextrin for treatment of kidney disease from L&F Research. S. M. holds equity interest in a company presently commercializing the form of cyclodextrin referenced in this paper. The patent associated with the use of hydroxypropyl-beta-cyclodextrin is published under US10,195,227. A.F. is Chief Medical Officer of LipoNexT, LLC. The remaining authors delare no competing interests.

