## [Peer Review File · Nature Communications]

Reviewers' comments:

Reviewer #1 (Remarks to the Author):

In their manuscript entitled “SMPDL3b is a Modulator of Insulin Receptor Signaling in Lipid Rafts: Implications for Diabetic Kidney Disease” by Mitrofanova et al., the authors investigate the role of the enzyme SMPDL3b in generation of bioactive lipid species with a focus on insulin receptor signaling and diabetes-induced kidney disease.

Using lipidomics analysis and enzymatic assays, the authors present evidence that overexpression of SMPDL3b leads to reduction of cellular ceramide-1-phosphate (C1P)—a class of bioactive lipids- and speculate that this reduction is a consequence of an as yet undescribed ceramide phosphatase activity of SMPDL3b. Furthermore, the authors find that SMPDL3b overexpression differentially affects insulin receptor (IR)-induced signaling and attribute this to altered IR surface expression and differential binding of caveolin-1 to IR isoforms A and B. In order to study the involvement of SMPDL3b in kidney biology in vivo, the authors further generated podocyte-specific knockout mice. While these mice were phenotypically normal under regular conditions, in a leptin receptor-deficient db/db background, conditional SMPDL3b deficiency protects the mice from several aspects of diabetic kidney disease and restores reduced C1P levels in kidney cortexes. Finally, administration of synthetic C1P was sufficient to remediate complications associated with DKD without negatively affecting kidney function.

In general, the paper by Mitrofanova et al. deals with an interesting subject matter, shedding light at the role of SMPDL3b and C1P in renal biology and disease. While I find the presented in vivo evidence for an implication of SMPDL3b in DKD convincing and the fact that diabetes-associated complications can be remediated by administration of its potential substrate C1P exciting, I am less convinced by the suggested mechanism of SMPDL3b function derived from the presented in vitro experiments.

Especially, based on the current experimental evidence, I am not convinced, that overexpressed SMPDL3b modulates AKT/mTOR via differential engagement of IR isoforms. Also the implication of C1P or SMPDL3b enzymatic activity in this process (if required) is not clear.

Major points:

1) Enzymatic activity of SMPDL3b towards C1P

While it seems clear that SMPDL3b affects cellular C1P abundance, it is difficult to clarify, if the observed effects on C1P levels in SMPDL3b overexpressing podocytes and db/db mice as well as the

increased conversion of C1P to ceramide in SMPDL3b expressing HEK293T cells is a direct enzymatic activity of SMPDL3b towards C1P or a circumstantial phenomenon. To further clarify this, the authors could include an enzymatic inactive form of the enzyme (e.g. the H135A mutant used in Fig. 3b) in the C6-NBD-C1P enzymatic assay – or even better – use purified (e.g. immunoprecipitated) proteins.

2) RNASeq experiment

When comparing transcriptional changes in SMPDL3b overexpressing vs. control podocytes, the authors report differential expression of 5182 mRNAs. From Supplemental Table 2 it appears that this number stems from filtering on bare P values (PValue) and not on FDR corrected p values (FDR), which in my opinion would be more appropriate to report on the significant changes. In any case, the number of affected mRNAs is large, indicating perturbation of many cellular processes by SMPDL3b overexpression. Instead of highlighting 29 selected genes in PI3K-AKT, InsSignal and mTOR pathways, it would be better to integrate all the data obtained, e.g. by DAVID pathway enrichment analysis. The focus on PI3K-AKT, InsSignal and mTOR pathways could still be argued based on the background literature in addition to the differentially regulated genes. The data could also shed light on additional processes to be investigated to understand SMPDL3b-associated phenotypes.

3) Effect of SMPDL3b on insulin receptor signaling

The authors observed reduced insulin-induced AKT phosphorylation vs. increased mTOR activation in SMPDL3b overexpressing cells as evidenced by increased p70S6K and 4EBP1 phosphorylation states (e.g. Fig. 2bc, Fig. S2a). Even though this result seems consistent, I dis advise to use cropped western blots (even when indicated) if it is to compare states (Fig. 2b, Fig S2a). Also when presenting quantification of western blot data, I would expect the representative western blot to be consistent with the quantification. In Fig. 2c for example, insulin stimulation of control cells seems to reduce p70S6K phosphorylation at 1 nM in the western blot, but no effect is seen in the quantification.

What should be done is to generate a more consistent set of blots of control and SMPDL3b overexpressing cells upon insulin stimulation for phospho-AKT, 4EBP1 and p70S6K and the respective total proteins from the same samples. Ideally this panel also includes phospho-S6, which might be a more robust readout for mTOR activation. I also encourage to look at the kinetics of signaling over time, as the dynamics of signaling events could be affected.

4) Proposed mechanism of SMPDL3b in insulin receptor signaling

a) The authors found in overexpression experiments that IR isoforms and SMPDL3b efficiently co-immunoprecipitate (Fig. 3a) and report similar behavior in a fully endogenous setup (Fig. 3d). While the data are clear in overexpression, I am not so convinced by the endogenous experiment: In the IR panel, the input fraction shows two distinct bands at ca. 200 kDa and 90 kDa, most likely

representing the IR precursor and mature beta chain. In the IP fraction however, the observed band seems to migrate somewhere in between. Please comment.

b) Based on the co-immunoprecipitation experiment presented in Fig. 3c – a key figure – the authors claim that SMPDL3b differentially affects binding of IRA/IRB to Caveolin-1. Here the authors immunoprecipitate FLAG-tagged IRA or IRB and monitor the co-immunoprecipitation of Caveolin-1 upon increasing doses of SMPDL3b. In the IP fractions the figure clearly shows similar appearance of co-immunoprecipitated Caveolin-1, but the recovery of the directly immunoprecipitated IR isoforms is different. This is confusing, as I would have expected to see efficient immunoprecipitation of the FLAG-tagged IR and modulated binding of Caveolin-1. My conclusion of this experiment would be that SMPDL3b modulates binding of the FLAG antibody to the epitope tag used in IP and the claimed differential interaction between IR and Caveolin-1 cannot be deduced.

c) In Fig. 3fg it is suggested that Methyl-beta-cyclodextrin (CD) alters fractionation behavior of IR and SMPDL3b, “214: This occurred in association with increased IR localization (Fig. 3f) and decreased SMPDL3b at the PM (Fig. 3g)”. While the ratios between IR and Na/K-ATPase in the quantifications would suggest this, I don’t think it is appropriate to use this ratio for a statement on PM localization of IR here. In the representative western blot, Fig. 2f, CD treatment seems to increase the levels of Na/K-ATPase in the PM fraction, while IR seems not to be much affected (if anything the opposite). It also appears that SMPDL3b overexpression leads to an increase in Na/K-ATPase. To determine an effect on IR abundance on the PM, IR-specific FACS analysis would be an option and more quantitative. Also confocal microscopy of IR, Caveolin-1 and SMPDL3b should be considered. In Fig. 3g, comparing control and SMPDL3b overexpressing cells, it appears that the degree of overexpression of SMPDL3b in podocytes is not very strong. It is also not clear if the protein is tagged and should therefore migrate differently than the endogenous protein. I suggest to include a western blot of the whole cell lysate to better evaluate this.

Minor comments:

- 1) Figure 4c and Fig. 4d right panel: The bar graphs look identical. Please correct in case this is a mistake during figure assembly.
- 2) Fig. 5: The graphs reporting mesangial expansion and KW/BW Ratio should be exchanged to correspond to the text references, Line 285 and 287.
- 3) In Fig. 6a, C1P seems to induce AKT phosphorylation also in absence of insulin in the western blot, but not in the quantification. Please comment.
- 4) Recombinant C1P should be replaced by synthetic C1P, e.g. Line 343 and 402

Reviewer #2 (Remarks to the Author):

This is a well written and interesting paper by Mitrofanova et al. that presents intriguing data suggesting that increased podocyte sphingomyelin phosphodiesterase acid-like 3b expression excess negatively affects the availability of ceramide-1-phosphate (C1P) leading to impaired insulin signaling and podocyte injury. The authors demonstrate by an elegant series of in vitro studies that SMPDL3b-mediated deficiency of C1P impairs insulin signaling by displacing insulin receptor isoform B from caveolin-1, a step that is essential for insulin signaling and action. In vivo, the authors demonstrate that podocyte-specific deletion of SMPDL3b in db/db mice reduces albuminuria, restores C1P levels and protein kinase B phosphorylation in kidney cortice. Most importantly, C1P replacement therapy in vivo is sufficient to ameliorate diabetic kidney disease (DKD), suggesting a potential role of SMPDL3b and C1P in the progression of DKD and offering a new lipid-based treatment option for patients with diabetic complications.

Although the data presented are highly innovative and significant, and a large number of in vitro and in vivo experiments are well designed to support the hypothesis, there are some concerns that should be addressed.

Major concerns:

1. The work relies almost exclusively on overexpression studies. No loss of function data are presented in cultured cells. How does knockout of SMPDL3b affect sphingolipid composition in podocytes?
2. The inclusion of SMPDL3b glomerular expression data in research kidney biopsies from patients affected by DKD would strengthen the paper.
3. The authors previously published that cyclodextrin protects from DKD. Does cyclodextrin restore the interaction of caveolin-1 with Insulin receptor B in SMPDL3b overexpressing cells?
4. Is podocyte specific SMPDL3b deficiency sufficient to influence C1P content in kidney cortex?

Minor concern:

1. Line 86: it would be more appropriate if the authors would change "C1P deficiency" to "a reduction of biologically active C1P". Similarly, C1P replacement therapy should be replaced by C1P supplementation therapy throughout the manuscript. In the present study, endogenous C1P is not replaced but exogenous C1P is administered to restore physiological C1P levels. Therefore, the term C1P replacement therapy is misleading.

Reviewer #3 (Remarks to the Author):

Mitrofanova and colleagues describe the role of SMPDL3b in pro-survival insulin signaling in podocytes. The authors demonstrate that SMPDL3b converts ceramide-1-P to ceramide. Podocyte-specific overexpression of SMPDL3b, which mimics the enhanced expression observed in the diabetic kidney, resulted in impaired insulin-dependent AKT signaling but augmented p70S6K phosphorylation. This was found to be the result of reduced IR localization to the plasma membrane and specific interference of the caveolin 1- IRB interaction. In vivo deletion of SMPDL3b specifically in podocytes resulted in mice that were protected from DKD and reduced podocyte loss. Furthermore, administration of C1P phenocopied the genetic SMPDL3b ablation model displaying enhanced insulin signaling, survival of podocytes and protection from DKD.

This is a well-designed study, and an example of a rare case where the study describes the process from the biochemical mechanism to a therapeutic approach. The in vivo studies demonstrate a particularly striking protection from DKD, which holds therapeutic promise for diabetic complications. Overall, this study provides important, new insights into the regulation of insulin signaling by sphingolipids and offers a new potential target for treatment of DKD. It therefore creates a high level of enthusiasm. The lipidomics work is excellent and all the metabolite levels (relative differences between subspecies) are within what is expected. No bias or artifacts should be expected from the methodology used.

However, there are a couple of issues that deserve more attention:

- 1) Is the inhibition of insulin signaling by SMPDL3b overexpression dependent on the depletion of the enzymes substrate (C1P) or the production of the enzyme's product (ceramides)? The authors cite papers from the Summers group who demonstrated the well-known role of ceramide species in insulin resistance. Although there is no build-up of ceramides in the SMP overexpressing cells by lipidomics, the local production of ceramides or ceramide degradation products at the level of the lipid raft may be exerting effects on insulin signaling, in addition to or instead of C1P. Is it possible to test this (like a C1P analogue that cannot be dephosphorylated)? The authors should minimally discuss the possibility or provide a case as to why they believe it is C1P not ceramides. Just pointing out that the C1P concentration is two orders of magnitude smaller than ceramides makes it unlikely of course, but it should be discussed.
- 2) Can the improvements in kidney function seen with C1P administration be due to improvements in overall systemic metabolism? In other words, does C1P make the mouse more insulin sensitive or glucose tolerant? Either way, whether this effect is specific to the kidney or occurs through improvements in whole body metabolism, C1P administration is still an interesting therapeutic strategy. However, for a better mechanistic understanding, it is important to test this.
- 3) There are several western blot figures where conditions that are assumed to be run on the same gel are cropped separately such as supplemental figure 2A or figure 2B. In these figures, we are directly comparing the gel lanes for expression but they are cropped separately. The authors

should clearly indicate in the figure legend why it was necessary to make the figure like that. Other similar examples of this are Figure 3 F and G or Figure 6 A and B.

4) The immunofluorescence images in Supplemental figures 4D and 5D are too dark on a computer screen or on a printed version. There is no staining visible.

February 1st, 2019

Dear Reviewers,

We appreciate the insightful and constructive comments provided by you about our study. We believe that we have addressed all comments and concerns raised and we have performed additional *in vitro* and *in vivo* experiments as requested.

Reviewer #1 (Remarks to the Author):

In their manuscript entitled "SMPDL3b is a Modulator of Insulin Receptor Signaling in Lipid Rafts: Implications for Diabetic Kidney Disease" by Mitrofanova et al., the authors investigate the role of the enzyme SMPDL3b in generation of bioactive lipid species with a focus on insulin receptor signaling and diabetes-induced kidney disease. Using lipidomics analysis and enzymatic assays, the authors present evidence that overexpression of SMPDL3b leads to reduction of cellular ceramide-1-phosphate (C1P)—a class of bioactive lipids- and speculate that this reduction is a consequence of an as yet undescribed ceramide phosphatase activity of SMPDL3b. Furthermore, the authors find that SMPDL3b overexpression differentially affects insulin receptor (IR)-induced signaling and attribute this to altered IR surface expression and differential binding of caveolin-1 to IR isoforms A and B. In order to study the involvement of SMPDL3b in kidney biology *in vivo*, the authors further generated podocyte-specific knockout mice. While these mice were phenotypically normal under regular conditions, in a leptin receptor-deficient db/db background, conditional SMPDL3b deficiency protects the mice from several aspects of diabetic kidney disease and restores reduced C1P levels in kidney cortexes. Finally, administration of synthetic C1P was sufficient to remediate complications associated with DKD without negatively affecting kidney function.

In general, the paper by Mitrofanova et al. deals with an interesting subject matter, shedding light at the role of SMPDL3b and C1P in renal biology and disease. While I find the presented *in vivo* evidence for an implication of SMPDL3b in DKD convincing and the fact that diabetes-associated complications can be remediated by administration of its potential substrate C1P exciting, I am less convinced by the suggested mechanism of SMPDL3b function derived from the presented *in vitro* experiments. Especially, based on the current experimental evidence, I am not convinced, that overexpressed SMPDL3b modulates AKT/mTOR via differential engagement of IR isoforms. Also the implication of C1P or SMPDL3b enzymatic activity in this process (if required) is not clear.

Major points:

1) Enzymatic activity of SMPDL3b towards C1P. While it seems clear that SMPDL3b affects cellular C1P abundance, it is difficult to clarify, if the observed effects on C1P levels in SMPDL3b overexpressing podocytes and db/db mice as well as the increased conversion of C1P to ceramide in SMPDL3b expressing

HEK293T cells is a direct enzymatic activity of SMPDL3b towards C1P or a circumstantial phenomenon. To further clarify this, the authors could include an enzymatic inactive form of the enzyme (e.g. the H135A mutant used in Fig. 3b) in the C6-NBD-C1P enzymatic assay – or even better – use purified (e.g. immunoprecipitated) proteins.

We thank the reviewer for this comment. As recommended, we performed a set of experiments using purified SMPDL3b protein loaded in different concentrations. Results of this experiment are added in Supplementary Figure 1 in the revised manuscript. Determination of SMPDL3b enzymatic activity with recombinant protein failed to demonstrate a direct phosphatase activity (Supplementary Figure 1b). While this could be due to the utilization of a soluble non GPI-anchored protein or to the absence of an essential cofactor, we can only conclude that SMPDL3b affects the conversion of C1P into ceramide and have modified the manuscript accordingly.

2) RNASeq experiment. When comparing transcriptional changes in SMPDL3b overexpressing vs. control podocytes, the authors report differential expression of 5182 mRNAs. From Supplemental Table 2 it appears that this number stems from filtering on bare P values (PValue) and not on FDR corrected p values (FDR), which in my opinion would be more appropriate to report on the significant changes. In any case, the number of affected mRNAs is large, indicating perturbation of many cellular processes by SMPDL3b overexpression. Instead of highlighting 29 selected genes in PI3K-AKT, InsSignal and mTOR pathways, it would be better to integrate all the data obtained, e.g. by DAVID pathway enrichment analysis. The focus on PI3K-AKT, InsSignal and mTOR pathways could still be argued based on the background literature in addition to the differentially regulated genes. The data could also shed light on additional processes to be investigated to understand SMPDL3b-associated phenotypes.

We thank the reviewer for this comment. As suggested by the reviewer, we filtered RNASeq results based on FDR corrected p values only, which resulted in differential expression of 1948 genes in SMPDL3b overexpressing podocytes. We have revised the manuscript, Figure 2 and Supplementary Table 2 accordingly. As recommended, using DAVID pathway enrichment analysis of the genes passed FDR correction, we found that 65 signaling pathways are affected in SMPDL3b overexpression cells. Figure 2a was updated to present pathways related to the current study only. Supplementary Table 2 was updated to present the all discovered pathways, not mRNA sequencing results.

3) Effect of SMPDL3b on insulin receptor signaling. The authors observed reduced insulin-induced AKT phosphorylation vs. increased mTOR activation in SMPDL3b overexpressing cells as evidenced by increased p70S6K and 4EBP1 phosphorylation states (e.g. Fig. 2bc, Fig. S2a). Even though this result seems consistent, I disadvice to use cropped western blots (even when indicated) if it is to compare states (Fig. 2b, Fig S2a). Also when presenting quantification of western blot data, I would expect the representative western blot to be consistent with the quantification. In Fig. 2c for example, insulin stimulation of control cells seems to reduce p70S6K phosphorylation at 1 nM in the western blot, but no effect is seen in the quantification. What should be done is to generate a more consistent set of blots of control and SMPDL3b overexpressing cells upon insulin stimulation for phospho-AKT, 4EBP1 and p70S6K and the respective total proteins from the same samples. Ideally this panel also includes phospho-S6, which might be a more robust readout for mTOR activation. I also encourage to look at the kinetics of signaling over time, as the dynamics of signaling events could be affected.

We thank the reviewer for this comment. As suggested, more consistent set of blots for pAKT/tAKT, p-p70S6K/t-p70S6K, and p-4EB-P1/t-4E-BP1 under insulin stimulation had been generated. Figure 2 has been updated accordingly. We have also added data on SMPDL3b knockout podocytes (siSMP) in this figure as suggested by Reviewer #2. Western blot pictures from old Figure 2b and Figure 2c were combined into one new Figure 2b, while related quantification presented on new Figure 2c. As suggested, we have performed experiments on kinetics of signaling over time (0-30 min) using 1nM insulin stimulation during 30 min (n=3). New data showing that CTRL and siSMP podocytes have similar kinetics of AKT, p70S6K and 4EB-P1 phosphorylation with significant changes in AKT phosphorylation only, while SMP OE cells have significant changes in p70S6K and 4EB-P1 phosphorylation, were included in the revised manuscript and presented on new Figure 2d. In addition, as suggested by the reviewer in the comment 4c, we have generated new set of IR-specific FACS experiments to determine an effect on IR abundance on the PM and included these data into new Figure 2g. Supplementary Figure 2 was revised accordingly.

4) Proposed mechanism of SMPDL3b in insulin receptor signaling.

a) The authors found in overexpression experiments that IR isoforms and SMPDL3b efficiently co-immunoprecipitate (Fig. 3a) and report similar behavior in a fully endogenous setup (Fig. 3d). While the data are clear in overexpression, I am not so convinced by the endogenous experiment: In the IR panel, the input fraction shows two distinct bands at ca. 200 kDa and 90 kDa, most likely representing the IR precursor and mature beta chain. In the IP fraction however, the observed band seems to migrate somewhere in between. Please comment.

We thank the reviewer for this remark. Experiments on endogenous IP were performed using the Dynabeads M-270 Epoxy Co-IP kit (Thermo Fisher). During the samples preparation buffers with different conditions were used. Dynabeads magnetic beads that are used for IP pull-down are non-compatible with any reducing agent as it affects the solubility of iron in beads making it a chelating agent and reducing the iron in the magnetic beads. Thus, for total lysates preparation a loading buffer with reducing agent (beta mercaptoethanol) and higher pH is suggested to use, while for IP eluates a loading buffer without reducing agent and lower pH is suggested to use. As an insulin receptor is a protein that has several subunits that linked by disulfide bonds, the molecular weight will be different in reducing (lower band) and non-reducing (higher band) conditions. This was confirmed by contacting a support technician from the company (Dr. Suleman Funmilayo, e-mail: ts.proteins@thermofisher.com; tel +1-800-955-6288, 2, 1, 2). We have repeated endogenous IP experiments according to the recommendations of the technician and Figure 3d was revised accordingly.

b) Based on the co-immunoprecipitation experiment presented in Fig. 3c – a key figure – the authors claim that SMPDL3b differentially affects binding of IRA/IRB to Caveolin-1. Here the authors immunoprecipitate FLAG-tagged IRA or IRB and monitor the co-immunoprecipitation of Caveolin-1 upon increasing doses of SMPDL3b. In the IP fractions the figure clearly shows similar appearance of co-immunoprecipitated Caveolin-1, but the recovery of the directly immunoprecipitated IR isoforms is different. This is confusing, as I would have expected to see efficient immunoprecipitation of the FLAG-tagged IR and modulated binding of Caveolin-1. My conclusion of this experiment would be that SMPDL3b modulates binding of the FLAG antibody to the epitope tag used in IP and the claimed differential interaction between IR and Caveolin-1 cannot be deduced.

We thank the reviewer for this important point. After careful reviewing of the original blots, we found that the

cropped blots indicating IP results were mislabeled. Please find the original blots' scans below. Figure 3c was revised accordingly.

c) In Fig. 3fg it is suggested that Methyl-beta-cyclodextrin (CD) alters fractionation behavior of IR and SMPDL3b, “214: This occurred in association with increased IR localization (Fig. 3f) and decreased SMPDL3b at the PM (Fig. 3g)”. While the ratios between IR and Na/K-ATPase in the quantifications would suggest this, I don’t think it is appropriate to use this ratio for a statement on PM localization of IR here. In the representative western blot, Fig. 2f, CD treatment seems to increase the levels of Na/K-ATPase in the PM fraction, while IR seems not to be much affected (if anything the opposite). It also appears that SMPDL3b overexpression leads to an increase in Na/K-ATPase. To determine an effect on IR abundance on the PM, IR-specific FACS analysis would be an option and more quantitative. Also confocal microscopy of IR, Caveolin-1 and SMPDL3b should be considered. In Fig. 3g, comparing control and SMPDL3b overexpressing cells, it appears that the degree of overexpression of SMPDL3b in podocytes is not very strong. It is also not clear if the protein is tagged and should therefore migrate differently than the endogenous protein. I suggest to include a western blot of the whole cell lysates to better evaluate this.

We thank the reviewer for this comment. To confirm that CD does not affect Na/K-ATPase localization at the plasma membrane, but does affect localization of the IR and SMPDL3b, we re-run Western blots, combined together old Figure 3g and 3f and displayed more consistent representative picture on new Figure 3g. As suggested by the reviewer, we performed FACS analysis to determine an effect on IR abundance on the plasma membrane in control and SMPDL3b overexpressing human podocytes and displayed data on new Figure 2g (please see comment 3 above). We previously have shown in total cell lysates that endogenous SMPDL3b band

appears at 50 kDa size¹, while GFP-tagged SMPDL3b corresponds to 70-75 kDa band (Fornoni et al, Science TM, 2011). While we agree with the reviewer that confocal microscopy of IR caveolin-1 and SMPDL3b would have further validated our findings, consistent with findings from others² endogenous IR is almost undetectable in human podocytes by immunofluorescence, and preliminary results did not show any consistent findings. We hope that the addition of the new FACS analysis is sufficient to address the reviewer suggestion.

Minor comments:

1) Figure 4c and Fig. 4d right panel: The bar graphs look identical. Please correct in case this is a mistake during figure assembly.

We thank the reviewer for pointing out this discrepancy. We have corrected figures accordingly.

2) Fig. 5: The graphs reporting mesangial expansion and KW/BW Ratio should be exchanged to correspond to the text references, Line 285 and 287.

We have modified the manuscript accordingly.

3) In Fig. 6a, C1P seems to induce AKT phosphorylation also in absence of insulin in the western blot, but not in the quantification. Please comment.

We thank the reviewer for pointing out this discrepancy. We have generated new set of more convinced blots showing restoration of AKT phosphorylation in SMP OE podocytes in response to C1P and insulin stimulation. New data are presented on new Figure 6a.

4) Recombinant C1P should be replaced by synthetic C1P, e.g. Line 343 and 402.

We have modified the manuscript accordingly.

Reviewer #2 (Remarks to the Author):

This is a well written and interesting paper by Mitrofanova et al. that presents intriguing data suggesting that increased podocyte sphingomyelin phosphodiesterase acid-like 3b expression excess negatively affects the availability of ceramide-1-phosphate (C1P) leading to impaired insulin signaling and podocyte injury. The authors demonstrate by an elegant series of in vitro studies that SMPDL3b-mediated deficiency of C1P impairs insulin signaling by displacing insulin receptor isoform B from caveolin-1, a step that is essential for insulin signaling and action. In vivo, the authors demonstrate that podocyte-specific deletion of SMPDL3b in db/db mice reduces albuminuria, restores C1P levels and protein kinase B phosphorylation in kidney cortice. Most importantly, C1P replacement therapy in vivo is sufficient to ameliorate diabetic kidney disease (DKD), suggesting a potential role of SMPDL3b and C1P in the progression of DKD and offering a new lipid-based treatment option for patients with diabetic complications. Although the data presented are highly innovative and significant, and a large number of in vitro and in vivo

experiments are well designed to support the hypothesis, there are some concerns that should be addressed.

Major concerns:

1) The work relies almost exclusively on overexpression studies. No loss of function data are presented in cultured cells. How does knockout of SMPDL3b affect sphingolipid composition in podocytes?

We thank the reviewer for this important comment. We have added data on SMPDL3b knockdown podocytes, including data on lipidomics (Figure 1a-d, Supplementary table 1), data on protein phosphorylation and expression of AKT, p70S6K, and 4EB-P1 (Figure 2b, c), data on insulin signaling kinetics (Figure 2d). SMPDL3b knockdown podocytes demonstrated no changes in total sphingomyelin or total ceramide lipid composition, while total amount of ceramide-1-phosphate was significantly decreased.

2) The inclusion of SMPDL3b glomerular expression data in research kidney biopsies from patients affected by DKD would strengthen the paper.

We thank the reviewer for this point. We previously reported (as show on the Figure below) that glomerular expression of SMPDL3b is elevated in 70 patients with DKD³.

3) The authors previously published that cyclodextrin protects from DKD. Does cyclodextrin restore the interaction of caveolin-1 with Insulin receptor B in SMPDL3b overexpressing cells?

We thank the reviewer for this important comment. To answer this question, we performed additional set of IP experiments in HE293 cells transfected with IR-FLAG, IRB-FLAG, Cav1-GFP, and SMPDL3b-HA constructs and treated with 5 mM CD for 1 hour. Obtained results confirmed that CD restores interaction between IRB and caveolin-1 and abrogates interaction between IRA and caveolin-1. These data were added to the manuscript and are reflected on Figure 3f.

4) Is podocyte specific SMPDL3b deficiency sufficient to influence C1P content in kidney cortex?

We thank the reviewer for this comment. We had performed additional LC-MS assay on kidney cortexes from

podocyte specific *Smpdl3b*-deficient mice, which demonstrated no significant changes in total sphingomyelin, total ceramide or total ceramide-1-phosphate across analyzed groups of mice. New data were incorporated into the manuscript as Supplementary Figure 3d-f.

Minor concern:

1) Line 86: it would be more appropriate if the authors would change “C1P deficiency” to “a reduction of biologically active C1P”. Similarly, C1P replacement therapy should be replaced by C1P supplementation therapy throughout the manuscript. In the present study, endogenous C1P is not replaced but exogenous C1P is administered to restore physiological C1P levels. Therefore, the term C1P replacement therapy is misleading.

We have modified the manuscript accordingly.

Reviewer #3 (Remarks to the Author):

Mitrofanova and colleagues describe the role of SMPDL3b in pro-survival insulin signaling in podocytes. The authors demonstrate that SMPDL3b converts ceramide-1-P to ceramide. Podocyte-specific overexpression of SMPDL3b, which mimics the enhanced expression observed in the diabetic kidney, resulted in impaired insulin-dependent AKT signaling but augmented p70S6K phosphorylation. This was found to be the result of reduced IR localization to the plasma membrane and specific interference of the caveolin 1- IRB interaction. In vivo deletion of SMPDL3b specifically in podocytes resulted in mice that were protected from DKD and reduced podocyte loss. Furthermore, administration of C1P phenocopied the genetic SMPDL3b ablation model displaying enhanced insulin signaling, survival of podocytes and protection from DKD.

This is a well-designed study, and an example of a rare case where the study describes the process from the biochemical mechanism to a therapeutic approach. The in vivo studies demonstrate a particularly striking protection from DKD, which holds therapeutic promise for diabetic complications. Overall, this study provides important, new insights into the regulation of insulin signaling by sphingolipids and offers a new potential target for treatment of DKD. It therefore creates a high level of enthusiasm. The lipidomics work is excellent and all the metabolite levels (relative differences between subspecies) are within what is expected. No bias or artifacts should be expected from the methodology used.

However, there are a couple of issues that deserve more attention:

1) Is the inhibition of insulin signaling by SMPDL3b overexpression dependent on the depletion of the enzymes substrate (C1P) or the production of the enzyme’s product (ceramides)? The authors cite papers from the Summers group who demonstrated the well-known role of ceramide species in insulin resistance. Although there is no build-up of ceramides in the SMP overexpressing cells by lipidomics, the local

production of ceramides or ceramide degradation products at the level of the lipid raft may be exerting effects on insulin signaling, in addition to or instead of C1P. Is it possible to test this (like a C1P analogue that cannot be dephosphorylated)? The authors should minimally discuss the possibility or provide a case as to why they believe it is C1P not ceramides. Just pointing out that the C1P concentration is two orders of magnitude smaller than ceramides makes it unlikely of course, but it should be discussed.

We thank the reviewer for this very important remark regarding the role of ceramide in insulin signaling. The point of the reviewer is well taken, as it is well known that ceramide is a major inducer of insulin resistance and it is possible that the increased ceramide generation in SMPDL3b OE cells still plays a key role in the induction of IR despite not being amenable to build-up because of its rapid degradation. The focus on C1P as a major player came from C1P replacement experiments *in vitro* and *in vivo*. Should ceramide rather than C1P deficiency be the major driver of insulin resistance in our model, then the administration of C1P in diabetes would result in increased ceramide production and further worsening of DKD, and the observed improvement of AKT phosphorylation *in vitro* and *in vivo* would remain unexplained. Furthermore, preliminary experiments performed with ASMase inhibitors to reduce ceramide production in podocytes were effective in reducing ceramide content but did not demonstrate restoration of insulin signaling (data not shown).

2) Can the improvements in kidney function seen with C1P administration be due to improvements in overall systemic metabolism? In other words, does C1P make the mouse more insulin sensitive or glucose tolerant? Either way, whether this effect is specific to the kidney or occurs through improvements in whole body metabolism, C1P administration is still an interesting therapeutic strategy. However, for a better mechanistic understanding, it is important to test this.

We thank the reviewer for this comment. We have repeated a new set of *in vivo* experiments on db/db mice treated with 30 mg/kg C1P and performed intraperitoneal glucose tolerance test (IPGTT) to evaluate if C1P treatment improves glycaemic control. New data showing that C1P treatment has no significant effect on fasting glucose and glucose tolerance tests are included in the manuscript and reflected on Figure 7g, h.

3) There are several western blot figures where conditions that are assumed to be run on the same gel are cropped separately such as supplemental figure 2A or figure 2B. In these figures, we are directly comparing the gel lanes for expression but they are cropped separately. The authors should clearly indicate in the figure legend why it was necessary to make the figure like that. Other similar examples of this are Figure 3 F and G or Figure 6 A and B.

We thank the reviewer for pointing out this discrepancy. All of the mentioned Western blots were run on the same gels. The reason why they were cropped separately is to remove empty vector cell line data for Figures 2a and 2b and to remove the marker line for Figures 3f and 3g and Figures 6a, which was loaded in between CTRL and SMP OE samples. The original blot scans are provided in the Figure below. Figure 6b shows uncropped blot. To avoid this confusing situation with other potential readers of this paper and in accordance with comments of other reviewers, we updated Figure 2a and b (Figure 2b and Figure 2c in new version), Figure 3f and g (Figure 3g in new version), Figure 6a and b (Figure 6a and b in new version).

4) The immunofluorescence images in Supplemental figures 4D and 5D are too dark on a computer screen or on a printed version. There is no staining visible.

We thank the reviewer for this comment. Images on Supplemental Figures 4d and 5d reflects negative controls for staining of AKT and Synaptopodin in mouse kidney biopsies. Since this information is not critical for the study's results understanding and raises questions, we removed those figures from new version of the manuscript.

Related references:

1. Fornoni A, *et al.* Rituximab targets podocytes in recurrent focal segmental glomerulosclerosis. *Science translational medicine* **3**, 85ra46 (2011).
2. Coward RJ, *et al.* The human glomerular podocyte is a novel target for insulin action. *Diabetes* **54**, 3095-3102 (2005).
3. Yoo TH, *et al.* Sphingomyelinase-like phosphodiesterase 3b expression levels determine podocyte injury phenotypes in glomerular disease. *Journal of the American Society of Nephrology : JASN* **26**, 133-147 (2015).

REVIEWERS' COMMENTS:

Reviewer #1 (Remarks to the Author):

The authors have answered all my points satisfactorily. I would suggest however to include the original table containing the RNASeq data on the gene level in addition to the current DAVID enrichment table, as this might be an interesting resource for other researchers. Furthermore, please adapt Fig. 3g, some labels of the western blots appear misaligned.

Reviewer #2 (Remarks to the Author):

The authors have addressed all my concerns and I don't have any more comments.

Reviewer #3 (Remarks to the Author):

The authors have adequately addressed the original concerns.

April 19th, 2019

Dear Reviewers,

We appreciate you taking the time to evaluate our manuscript entitled "SMPDL3b Modulates Insulin Receptor Signaling in Diabetic Kidney Disease" once more. We believe that we addressed all additional concerns raised.

Reviewer #1 (Remarks to the Author):

The authors have answered all my points satisfactorily. I would suggest however to include the original table containing the RNASeq data on the gene level in addition to the current DAVID enrichment table, as this might be an interesting resource for other researchers. Furthermore, please adapt Fig. 3g, some labels of the western blots appear misaligned.

Thank you for your suggestions. The original table containing the RNAseq data was added as Supplementary Data 3. Additionally, all raw data were deposited in the National Center for Biotechnology Information Gene Expression Omnibus (GEO) and are accessible through the GEO Series accession number GSE129666. Labels in Fig. 3g were aligned.

Reviewer #2 (Remarks to the Author):

The authors have addressed all my concerns and I don't have any more comments. We thank you for your useful comments which were incorporated into the manuscript thus solidifying our conclusions.

Reviewer #3 (Remarks to the Author):

The authors have adequately addressed the original concerns.

Thank you for your valuable feedback which greatly improved our manuscript.

Sincerely,

Alessia Fornoni MD PhD
Professor of Medicine
Chief, Katz Family Division of Nephrology and Hypertension
Director, Peggy and Harold Katz Family Drug Discovery Center Batchelor Building 6th floor
1580 NW 10th Ave, Miami, FL, 33136
E-mail: afornoni@med.miami.edu
Assistant: Janessa Maidment
E-mail: jxg1536@med.miami.edu
Tel: +1-305-243-7745 <http://medicine.med.miami.edu/nephrology>